

# Five-year records of Total Mercury Deposition flux at GMOS sites in the Northern and Southern Hemispheres

Francesca Sprovieri[1], Nicola Pirrone[2], Mariantonia Bencardino[1], Francesco D'Amore[1], Helene Angot[3,4], Carlo Barbante[17,10], Ernst-Günther Brunke[5], Flor Arcega-Cabrera[15], Warren Cairns[17], Sara Comero[8], María del Carmen Diéguez[7], Aurélien Dommergue[3,4], Ralf Ebinghaus[6], Xin Bin Feng[12], Xuewu Fu[12], Patricia Elizabeth Garcia[7], Bernd Manfred Gawlik[8], Ulla Hageström[9], Katarina Hansson[9], Milena Horvat[11], Jože Kotnik[11], Casper Labuschagne[5], Olivier Magand[4,3], Lynwill Martin[5], Nikolay Mashyanov[13], Thumeka Mkololo[5], John Munthe[9], Vladimir Obolkin[16], Martha Ramirez Islas[14], Fabrizio Sena[8], Vernon Somerset[5], Pia Spandow[9], Massimiliano Vardè[1,17], Chavon Walters[5], Ingvar Wängberg[9], Andreas Weigelt[6], Xu Yang[12], and Hui Zhang[12]

[1]CNR Institute of Atmospheric Pollution Research, Rende, Italy
[2]CNR Institute of Atmospheric Pollution Research, Rome, Italy
[3]Univ. Grenoble Alpes, Laboratoire de Glaciologie et Géophysique de l'Environnement, Grenoble, France
[4]CNRS, Laboratoire de Glaciologie et Géophysique de l'Environnement, Grenoble, France
[5]Cape Point GAW Station, Climate and Environ. Research & Monitoring, South African Weather Service
[6]Helmholtz-Zentrum Geesthacht, Germany
[7]INIBIOMA-CONICET-UNComa, Bariloche, Argentina
[8]Joint Research Centre, Italy
[9]IVL, Swedish Environmental Research Inst. Ltd., Sweeden
[10]University Ca' Foscari of Venice, Italy
[11]Jožef Stefan Institute, Lubliana, Slovenia
[12]State Key Laboratory of Environmental Geochemistry, Inst. of Geochemistry, Chinese Academy of Sciences
[13]St. Petersburg State University, Russia
[14]Instituto Nacional de Ecología y Cambio Climático (INECC), Mexico
[15]Universidad Nacional Autónoma de México (UNAM), Unidad de Química, Sisal, Mexico
[16]Limnological Institute (LIN), Siberian Branch of the Russian Academy of Sciences (SB RAS), Russia
[17]CNR Institute for the Dynamics of Environmental Processes, Venice, Italy

*Correspondence to:* Francesca Sprovieri (f.sprovieri@iia.cnr.it)

**Abstract.** The atmospheric deposition of mercury (Hg) occurs via several mechanisms including dry and wet scavenging by precipitation events. In an effort to understand the atmospheric cycling and seasonal depositional characteristics of Hg, wet deposition samples were collected for approximately five years at 17 selected GMOS monitoring sites located in the Northern and Southern Hemispheres in the framework of the Global Mercury Observation System (GMOS) project. Total mercury (THg)

5 exhibited annual and seasonal patterns in Hg wet deposition samples. Inter-annual differences in total wet deposition are mostly linked with precipitation volume, with the greatest deposition flux occurring in the wettest years. This data set provides a new insight into baseline concentrations of THg concentrations in precipitation worldwide, particularly in regions, such as the Southern Hemisphere and tropical areas where wet deposition as well as atmospheric Hg species were not investigated before, opening the way for future and additional simultaneous measurements across the GMOS network as well as new findings in

10 future modeling studies.





# 1 Introduction

Mercury (Hg) is a persistent pollutant of global concern due to its toxicity and its capacity to bioaccumulate aquatic food chains with serious consequences on human and wildlife health (Driscoll et al., 2013). Long-range atmospheric transport is the main pathway for contamination of remote ecosystems, therefore atmospheric deposition is the primary indicator for the understanding of its impact on aquatic and terrestrial ecosystems (Schroeder and Munthe, 1998; Lindberg et al., 2002). Hg exists in the atmosphere mainly in three operationally defined forms: gaseous elemental mercury (GEM),oxidized gaseous mercury (GOM), and particulate bound mercury (PBM). Globally, GEM is the predominant form whereas GOM and PBM are thought to be rapidly dry deposited and wet scavenged by precipitation (Lindberg et al., 2007). Due to the current lack of existing direct and accurate measurements of Hg dry deposition (Gustin et al., 2012; Zhang et al., 2012), the investigation of Hg fluxes to terrestrial and aquatic surfaces in different part of the world are mainly performed by wet deposition measurements (Gratz et al., 2009; Feng et al., 2009). Hg wet deposition represents the air-to-surface flux in precipitation (Lindberg et al., 2007). Previous studies suggested that the magnitude of Hg wet deposition varies geographically and seasonally due to climatic conditions, atmospheric chemistry, and human influences i.e. emissions of Hg from anthropogenic sources (Vanarsdale et al., 2005; Selin and Jacob, 2008; Prestbo and Gay, 2009). Current annual atmospheric deposition of Hg has been estimated to be 3200 $Mg\,y^{-1}$ deposited on land and 3700 $Mg\,y^{-1}$ into oceans (Mason et al., 2012). The preindustrial deposition rate has been estimated to be 1000 $Mg\,y^{-1}$ deposited on land and 2500 $Mg\,y^{-1}$ into oceans (Selin, 2009). Developed countries in North America and Europe have reduced their anthropogenic Hg use and emissions (Hylander, 2001), but Hg use and emission are still occurring widely around the world (Pacyna et al., 2010; Pirrone et al., 2010). In North America seasonal patterns in wet deposition are observed in both depositional flux and concentration with the highest values in summer and lowest values in winter (Pacyna et al., 2010; Mason et al., 2000);(Keeler et al., 2005; Choi et al., 2008; Prestbo and Gay, 2009). Explanations for this observation include more effective Hg scavenging by rain compared to snow (Keeler et al., 2005; Selin and Jacob, 2008), and a greater availability of soluble Hg due to convective transport in summer events (Keeler et al., 2005; Strode et al., 2007, 2008). Geographic differences in Hg wet deposition may be explained in part by the proximity to atmospheric sources. Results from the National Atmospheric Deposition Program's (NADP) Mercury Deposition Network (MDN) sites in the Northeastern United States exhibit a geographic trend with southern and coastal sites receiving higher Hg concentrations and depositional fluxes (Vanarsdale et al., 2005; Prestbo and Gay, 2009) due to their location nearer to the East coast megalopolis and downwind of anthropogenic emission sources such as coal burning power plants and waste incinerators. In addition, gaseous evasion of Hg from marine waters is a significant global source of atmospheric Hg and may also contribute to elevated depositional fluxes in coastal regions (Mason and Sheu, 2002). A similar pattern exists in northern Europe with a clear gradient in atmospheric concentrations and deposition (Munthe et al., 2003). Hg wet deposition data are therefore important for verifying atmospheric models, understanding the biogeochemical cycling of Hg on a regional/global scale, and investigating ecosystem impacts. Regional monitoring networks with properly chosen monitoring sites can provide accurate estimates of wet deposition at regional scales. Long-term Hg wet deposition measurements exist at many locations within the United States as part of the MDN or in Europe as part of the EMEP program; however, before the establishment of the global Hg network by the GMOS,



**Table 1.** Key information on the 17 GMOS monitoring sites

| | | Code | Name | Country | Lat | Lon | Elev. | Collector Type | Type* |
|---|---|---|---|---|---|---|---|---|---|
| Northern Hemisphere | 1 | NYA | Ny-Ålesund | Norway | 78,90 | 11,88 | 12 | bulk-modified | M |
| | 2 | PAL | Pallas | Finland | 68,00 | 24,24 | 340 | bulk-modified | S |
| | 3 | RAO | Råö | Sweden | 57,39 | 11,91 | 5 | bulk-modified | M |
| | 4 | MHE | Mace Head | Ireland | 53,33 | -9,91 | 5 | wet-only | S |
| | 5 | LIS | Listvyanka | Russia | 51,85 | 104,89 | 670 | wet-only | S |
| | 6 | CMA | Col Margherita | Italy | 46,37 | 11,79 | 2545 | bulk-modified | S |
| | 7 | ISK | Iskrba | Slovenia | 45,56 | 14,86 | 520 | wet-only | M |
| | 8 | MCH | Mt. Changbai | China | 42,40 | 128,11 | 741 | wet-only | M/S |
| | 9 | LON | Longobucco | Italy | 39,39 | 16,61 | 1379 | wet-only | M |
| | 10 | MWA | Mt. Waliguan | China | 36,29 | 100,90 | 3816 | wet-only | M |
| | 11 | MAL | Mt. Ailao | China | 24,54 | 101,03 | 2503 | wet-only | S/M |
| Tropics | 12 | SIS | Sisal | Mexico | 21,16 | -90,05 | 7 | wet-only | S |
| | 13 | CST | Celestún | Mexico | 20,86 | -90,38 | 3 | wet-only | S |
| Southern Hemisphere | 14 | AMS | Amsterdam Island | TAAF | -37,80 | 77,55 | 70 | wet-only | M |
| | 15 | CPT | Cape Point | South Africa | -34,35 | 18,49 | 230 | wet-only | S |
| | 16 | CGR | Cape Grim | Australia | -40,68 | 144,69 | 94 | bulk-modified | S |
| | 17 | BAR | Bariloche | Argentina | -41,13 | -71,42 | 801 | wet-only | M |

* M=Master; S= Secondary

long-term of ambient Hg concentrations and measurements of Hg wet deposition fluxes were lacking (Lindberg et al., 2007; Selin, 2009; Zhang and Wright, 2009). Although a number of monitoring stations have been established to better understand the impact of Hg wet deposition on ecosystems in many countries in the Northern Hemisphere (Wängberg et al., 2007; Prestbo and Gay, 2009; Sanei et al., 2010) several regions of the world (i.e., regions which are becoming increasingly impacted by

5  anthropogenic activities in general), and prevalently the Tropical zone and the Southern Hemisphere, were lacking in wet deposition data available, in terms of concentrations and deposition Hg fluxes.

To address this concern, seasonal and annual variations of Hg wet deposition and concentration at 17 ground-based sites in the Northern and Southern Hemispheres were monitored as a part of GMOS (www.gmos.eu). Here an overview of the seasonal/annual Hg wet deposition patterns across the 17 sites, is presented, briefly examining meteorological/climatological

10  conditions, as well as indicators of anthropogenic air mass sources and/or atmospheric chemical conditions in relation to Hg wet deposition results observed. This study is the first multi-year comparison of Hg wet deposition worldwide and provides insights into annual and seasonal variations, as well as spatial gradient in Hg deposition patterns.





**Table 2.** Annual wet deposition flux $[\mu g m^{-2} y r^{-1}]$, cumulative rainfall amounts [mm], number of sampling days [d], weighted THg concentrations $[ng L^{-1}]$ and average wet deposition flux normalized to the number of sampling days $[ng m^{-2} d^{-1}]$ observed at the 17 GMOS ground-based monitoring sites for 2011 and 2012. Measures in bold are related to the calculations based on a restricted number of sampling days, therefore statistically less representative than the others

| | | 2011 | | | | | 2012 | | | | |
|---|---|---|---|---|---|---|---|---|---|---|---|
| | | Annual Wet Dep. Flux $[\mu g m^{-2} y r^{-1}]$ | Rainfall [mm] | ndays [d] | Weighted HgT $[ng L^{-1}]$ | Aver. Wet Dep. Flux $[ng m^{-2} d^{-1}]$ | Annual Wet Dep. Flux $[\mu g m^{-2} y r^{-1}]$ | Rainfall [mm] | ndays [d] | Weighted HgT $[ng L^{-1}]$ | Aver. Wet Dep. Flux $[ng m^{-2} d^{-1}]$ |
| Northern Hemisphere | NYA | - | - | - | - | - | 0,9 | 238,6 | 350 | 3,8 | 2,6 |
| | PAL | 2,9 | 407,4 | 363 | 7,1 | 8,0 | 1,9 | 278,6 | 332 | 6,8 | 5,7 |
| | RAO | 5,8 | 646,6 | 364 | 8,9 | 15,8 | 6,5 | 621,8 | 366 | 10,4 | 17,8 |
| | MHE | - | - | - | - | - | 0,9 | 393,7 | 113 | 2,2 | 7,6 |
| | LIS | - | - | - | - | - | **0,2** | **17,4** | **18** | **9,7** | **9,4** |
| | CMA | - | - | - | - | - | - | - | - | - | - |
| | ISK | 5,1 | 680,2 | 224 | 7,5 | 22,7 | 8,4 | 1349,7 | 363 | 6,2 | 23,2 |
| | MCH | 2,8 | 264,6 | 119 | 10,6 | 23,6 | 4,8 | 569,4 | 228 | 8,4 | 21,1 |
| | LON | - | - | - | - | - | **0,3** | **88,2** | **19** | **3,9** | **18,1** |
| | MWA | - | - | - | - | - | 0,3 | 79,5 | 127 | 4,3 | 2,7 |
| | MAL | 4,3 | 1543,2 | 222 | 2,8 | 19,5 | 3,2 | 971,5 | 202 | 3,3 | 16,1 |
| Tropics | SIS | - | - | - | - | - | - | - | - | - | - |
| | CST | - | - | - | - | - | 2,4 | 297,1 | 155 | 8,1 | 15,5 |
| Southern Hemisphere | AMS | - | - | - | - | - | - | - | - | - | - |
| | CPT | 0,3 | 133,5 | 119 | 2,1 | 2,4 | 3,8 | 260,3 | 147 | 14,6 | 25,8 |
| | CGR | - | - | - | - | - | - | - | - | - | - |
| | BAR | - | - | - | - | - | - | - | - | - | - |

**Table 3.** Annual wet deposition flux $[\mu g m^{-2} y r^{-1}]$, cumulative rainfall amounts [mm], number of sampling days [d], weighted THg concentrations $[ng L^{-1}]$ and average wet deposition flux normalized to the number of sampling days $[ng m^{-2} d^{-1}]$ observed at the 17 GMOS ground-based monitoring sites for 2013, 2014 and 2015. Measures in bold are related to the calculations based on a restricted number of sampling days, therefore statistically less representative than the others

| | | 2013 | | | | | 2014 | | | | | 2015 | | | | |
|---|---|---|---|---|---|---|---|---|---|---|---|---|---|---|---|---|
| | | Annual Wet Dep. Flux [*] | Rainfall [mm] | ndays [d] | Weighted HgT $[ng L^{-1}]$ | Aver. Wet Dep. Flux [**] | Annual Wet Dep. Flux [*] | Rainfall [mm] | ndays [d] | Weighted HgT $[ng L^{-1}]$ | Aver. Wet Dep. Flux [**] | Annual Wet Dep. Flux [*] | Rainfall [mm] | ndays [d] | Weighted HgT $[ng L^{-1}]$ | Aver. Wet Dep. Flux [**] |
| Northern Hemisphere | NYA | 0,9 | 225,4 | 243 | 4,1 | 3,8 | 1,7 | 293,3 | 357 | 5,7 | 4,7 | 0,8 | 171,7 | 180 | 4,4 | 4,2 |
| | PAL | 1,3 | 298,1 | 368 | 4,5 | 3,6 | 2,3 | 379,1 | 353 | 6,1 | 6,5 | - | - | - | - | - |
| | RAO | 4,2 | 515,2 | 365 | 8,2 | 11,5 | 6,3 | 631,6 | 365 | 9,9 | 17,2 | - | - | - | - | - |
| | MHE | 4,8 | 1048,8 | 363 | 8,2 | 13,3 | 4,1 | 623,3 | 119 | 6,6 | 34,7 | - | - | - | - | - |
| | LIS | **0,1** | **47,5** | **8** | **2,6** | **15,6** | - | - | - | - | - | - | - | - | - | - |
| | CMA | - | - | - | - | - | 4,4 | 559,5 | 219 | 7,8 | 20,0 | - | - | - | - | - |
| | ISK | 7,2 | 1364,4 | 350 | 5,3 | 20,6 | 10,0 | 1631,1 | 350 | 6,1 | 28,6 | 3,0 | 991,8 | 330 | 3,0 | 9,1 |
| | MCH | 1,2 | 300,4 | 121 | 3,9 | 9,6 | **1,0** | **177,0** | **85** | **5,4** | **11,3** | - | - | - | - | - |
| | LON | 3,1 | 472,6 | 208 | 6,6 | 15,0 | - | - | - | - | - | - | - | - | - | - |
| | MWA | 0,4 | 60,0 | 146 | 6,4 | 2,6 | **2,2** | **144,9** | **93** | **15,0** | **23,3** | - | - | - | - | - |
| | MAL | 5,5 | 1042,0 | 289 | 5,3 | 19,2 | **0,2** | **30,0** | **66** | **6,7** | **3,0** | - | - | - | - | - |
| Tropics | SIS | 7,4 | 669,6 | 361 | 11,0 | 20,5 | 6,5 | 712,5 | 368 | 9,1 | 17,7 | - | - | - | - | - |
| | CST | **0,1** | **6,2** | **13** | **13,5** | **6,5** | - | - | - | - | - | - | - | - | - | - |
| Southern Hemisphere | AMS | 1,95 | 833,2 | 272 | 2,34 | 7,2 | 1,55 | 864,1 | 328 | 1,80 | 4,7 | - | - | - | - | - |
| | CPT | 5,2 | 264,9 | 140 | 19,6 | 37,1 | 0,57 | 310,4 | 133 | 1,84 | 5,8 | 0,6 | 216,9 | 98 | 3,0 | 6,6 |
| | CGR | 3,1 | 775,6 | 290 | 4,0 | 10,6 | 3,8 | 562,3 | 337 | 6,7 | 11,2 | 3,1 | 477,4 | 247 | 6,5 | 12,6 |
| | BAR | - | - | - | - | - | **0,1** | **258,6** | **91** | **0,4** | **1,1** | 0,5 | 840,3 | 169 | 0,6 | 3,0 |

*uom: $[\mu g m^{-2} y r^{-1}]$

**uom: $[ng m^{-2} d^{-1}]$



## 2 Experimental

### 2.1 GMOS ground-based monitoring sites

The global Hg monitoring network has been established in the framework of the GMOS and presented in (Sprovieri et al., 2016). It has been developed by integrating previously on-going ground-based Hg monitoring stations as part of regional

networks with those established as part of GMOS also in regions of the world where atmospheric Hg measurements were previously limited. To date the GMOS network consists of 43 monitoring stations worldwide distributed and located in climatically diverse regions, including polar areas (Sprovieri et al., 2016). In the present study we refer the discussion on Hg wet deposition to a representative number of 17 ground-based sites distributed in the Northern and Southern Hemispheres. Table 1 provides key information on the 17 monitoring sites such as, their location (i.e., Country, coordinates etc.), elevation (m. asl)

and type of monitoring stations, Master and Secondary sites in respect to the atmospheric Hg measurements performed (Hg speciation and TGM/GEM measurements, respectively) along with THg wet deposition sampling.

### 2.2 Sample collection, analytical procedure, and QA/QC

Precipitation samples were collected across the sites primarily using wet-only collectors, (i.e., N-CON MDN or the Eigenbrodt NSA 171 wet-only samplers). Where necessary, due to site constraints or operator availability, few GMOS sites (Table 1) alter-

natively collected bulk precipitation samples. Within GMOS special attention was paid in respect to protocols harmonization, data quality collection and data management in order to assure a full comparability of site specific observational datasets. During the implementation stage of the GMOS global network, harmonized Standard Operating Procedures (SOPs) as well as common Quality Assurance/Quality Control (QA/QC) protocols have been addressed (Munthe et al., 2011) in accordance with the measurement practice adopted in well-established regional monitoring networks and based on the most recent liter-

ature (Brown et al., 2010a, b; Steffen et al., 2012; Gay et al., 2013). For THg in precipitation an ad-hoc Standard Operating Procedure has been developed and adopted within the network, and furthermore the management of the measurement program at most of the GMOS sites consisting in analysis of all precipitation samples, cleaning procedures, distribution of the sample bottles to all sites, have been performed by three reference laboratories (IVL, Sweden; CNR-IIA, Italy, and IJS, Slovenia) whereas the precipitation samples related to some other GMOS sites, in Russia (Listvyanka), in China (Mt. Walinguan, Mt.

Ailao, and Mt. Changbai), and in South Africa (Cape Point) have been analyzed by local laboratories. The analytical performance and the QA/QC of the analysis carried out by the reference laboratories as well as by the local laboratories were confirmed by the results achieved during International Inter-comparison exercises for Hg in water (i.e., Brooks Rand Instruments Inter-laboratory Comparison Study). GMOS sites predominantly collected bi-weekly samples. However, considering the spatial distribution and the diversity of meteorological parameters and conditions characterizing the monitoring sites locations,

the sampling frequency was sometime different across the sites. THg concentrations in precipitation samples, refrigerated and kept in the dark before the analysis (to avoid photo-induced reduction of the Hg in the precipitation sample), were determined according to the U.S. EPA Method 1631 (version E) (1631, 2002): each sample was first oxidized by BrCl (0.5 mL/100 mL sample), followed by neutralization with hydroxylamine hydrochloride ($NH_2OH \cdot HCl$). Stanneous chloride ($SnCl_2$) was




then added to the sample to reduce $Hg^{2+}_{(aq)}$ to $Hg^0_{(g)}$ which was quantified by Cold Vapor Atomic Fluorescence Spectrometry (CVAFS) using a Tekran Mercury Analysis System Model 2600 (Tekran Inc. Corporation, Canada). Working Hg standards solutions were obtained from a Standard Reference Material (SRM) produced by accredited laboratory (ISO/IEC 17025). Calibration standards were analyzed in the range from 0.2 to 100 ng/L (Recovery 93-109%). The standard curve was used within

the coefficient of determination ($r^2$) greater than 0.998 (linear). Initial (IPR) and ongoing precision and recovery (OPR) solutions (5 ppt) were analyzed prior to the analysis of samples and again after every 12 samples (Recovery 91-103%). These values were within the quality control acceptance criteria for performance in the EPA Method 1631e. The method detection limit (MDL; 40 CFR 136, Appendix B) for Hg has been determined to be 0.02 ng/L. The minimum level of quantification (ML) has been established as 0.05 ng/L for THg. The QA/QC of the analysis were obtained using replicates, method blanks,

field blanks, initial/ongoing precision recovery (IPR/OPR) standards, matrix spikes and certified reference materials (CRMs) with different certified Hg concentrations. Method and field blanks were always below the respective MDL, indicating minimal contamination during sampling, transport, and treatment for this study. Additionally, the sampling train materials [i.e., fluorinated polyethylene (FLPE) bottles, cylindrical glass funnels, Teflon adapters along with the glass capillary S-shaped tubes (to prevent loss of mercury from the sample) etc.] were thoroughly acid-cleaned and rinsed with ultra-pure water in the Hg

laboratory before and after sampling steps, and randomly tested for Hg concentrations; they were always below the MDL. All of these materials have been triple-bagged in zip-type plastic bags to keep them clean prior to use in the field. The results of "blanks" analysis allowed us to exclude possible contamination of all samples during different steps.

### 2.3 Hg wet deposition flux calculation

Considering the geographical distribution of the 17 sites located at different latitude and longitude, and therefore, under dif-

ferent meteorological and climatologically conditions, the precipitation was not collected over an entire year at each station due to limited amount of precipitation samples occurring during specific periods (i.e., dry seasons). Therefore, Hg flux was necessarily estimated based on the volume-weighted mean (VWM) concentration and the annual total precipitation amount collected at each site. The annual THg wet deposition flux can be approximated by the following equation:

$$F_W = C_{H_{gx}} \sum_{i=1}^{i=n} P^i 1/1000$$

where $F_W$ is the annual THg wet deposition flux ($\mu g m^{-2} yr^{-1}$), and $C_{H_{gx}}$ is the volume-weighted mean (VWM) concentration of THg ($ng L^{-1}$). $P^i (mm; 1mm = 1Lm^{-2})$ represents the precipitation amount associated to each wet deposition sample.

### 3 Hg wet deposition patterns and inter-annual variability

The annual variations in THg concentration and wet deposition recorded at all 17 monitoring GMOS sites are summarized in

Tables 2 and 3. Tables 2 and 3 list the monitoring sites according to their latitude and for each site, rain amounts collected, the number of the sampling days as well as the annual wet deposition flux and average THg wet deposition flux calculated for the period 2011-2015. The latter was calculated taking into account the number of sampling days at each site for each sample.





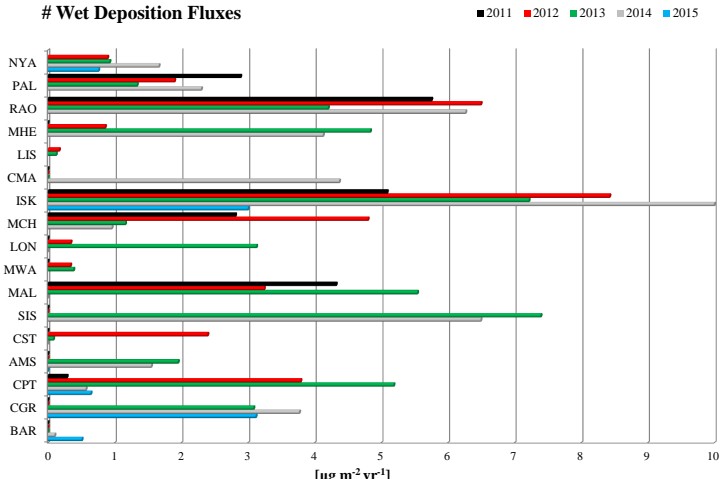

**Figure 1.** Annual THg wet deposition flux ($\mu g m^{-2} yr^{-1}$) during $2011 - 2015$ at the 17 GMOS sites

Annual THg wet deposition fluxes are shown in Figure 1.The Hg deposition at each site tends to vary from year to year, but to a different degree at different locations. It is well known that the magnitude of Hg wet deposition varies geographically and seasonally due to different meteorological and climatic conditions, atmospheric chemistry, and anthropogenic influences (Vanarsdale et al., 2005; Selin and Jacob, 2008; Prestbo and Gay, 2009). Therefore, considering the 10 sites distributed in the Northern Hemisphere, the discussion of the results will be separately related to the seven European sites and the three Chinese sites (see Tables 2 and 3) as well as those located in the tropical area and the sites distributed in the Southern Hemisphere. Considering the THg wet deposition from 2012 to 2014 at the European sites, there appears to be a geographical trend with an increase in Hg deposition from north (Arctic area, i.e., Ny Alesund, Pallas etc.) to south in the Northern Hemisphere (i.e., Rao, Mace Head, Listvyanka, Col Margherita, Longobucco). At the Chinese sites as well as at lower latitude (i.e., Tropical area and Southern Hemisphere) no spatial trend has been observed. However, it is important to point out that the sites in the Southern Hemisphere are limited in number compared to those in the Northern Hemisphere and the data coverage is less complete for each year considered. This makes detailed evaluation of spatial trends at the southern sites difficult. In addition, apart from Cape Point (CPT), no historical records of THg deposition exist for the new stations established in the GMOS project.

The geographical trend observed at the European stations with higher deposition of Hg in southern sites than in the north is in line with emission patterns with the main source areas in central and eastern Europe. The present data in combination with ground-based atmospheric Hg measurements performed within the GMOS project during 2012 - 2015 period indicate that these findings are in good agreement with the geographical distribution of atmospheric Hg with a downward gradient from the Northern to the Southern Hemisphere (Sprovieri et al., 2016). Figure 1 shows from 2012 to 2014 (the period with




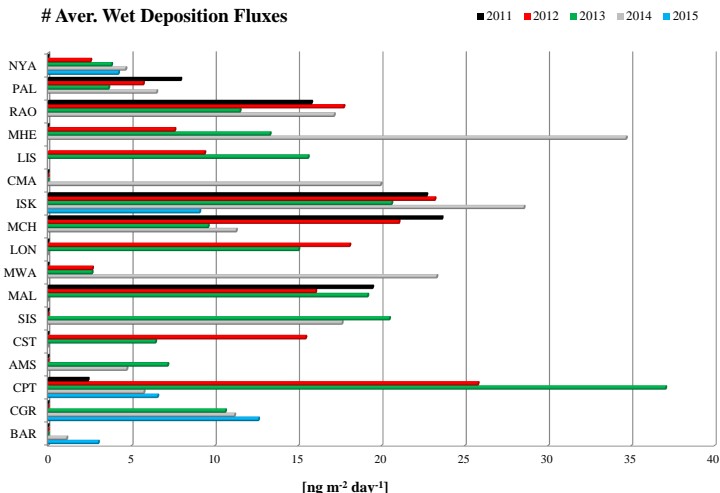

**Figure 2.** Average THg wet deposition Flux ($\mu g m^{-2} d^{-1}$) calculated during 2011-2015 at the 17 GMOS sites

more data coverage) a general increasing of THg wet deposition from Ny Alesund station (Norway) to Iskrba; this finding is particularly evident during the 2013 for sites at lower latitudes (i.e., Mace Head, Ireland, Col Margherita and Longobucco, Italy). This patterns is not apparent for other sites such as Listvyanka (Russia) indicating the influence of other emission sources or atmospheric transport pathways. In order to compare THg wet deposition at all sites and look for a confirmed geographical

trend in Europe, average wet deposition values were calculated ($ng m^{-2} d^{-1}$) normalizing the calculations on the effective number of sampling days. The results are shown in Figure 2. Comparing annual average wet deposition flux as is shown in Figure 1, and considering for example the 2013 period common to most of European sites, all measurements performed in the Northern Hemisphere, apart Col Margherita, where data is missing for that period, generally fits into a clear south to north decreasing trend. Deposition of atmospheric Hg at any given location is influenced by factors such as: (a) atmospheric Hg

concentration depending upon the local, regional and global sources; (b) site location in relation to the predominant wind direction in relation to the source areas; (c) precipitation amount which removes Hg from the atmosphere, and (d) length of precipitation events which affect Hg concentrations.

   In particular, Hg concentrations appear to be higher at the beginning of a precipitation event (i.e., rain or snow), and lower at the end of a precipitation event (Keeler et al., 2005; Gratz et al., 2009; Prestbo and Gay, 2009; Chen et al., 2014). This

is most evident during periods of prolonged precipitation (i.e., over a period of several days). It is obvious therefore that the Hg deposition obtained at some sites, is strongly influenced by the precipitation amounts. The annual deposition amounts during the 2011-2015 period is reported in Figure 3 which shows the influence of the precipitation amount on Hg deposition between, for example, Rao (Sweden) site and Pallas (Finland) site. The THg wet deposition fluxes recorded during 2011, 2012,




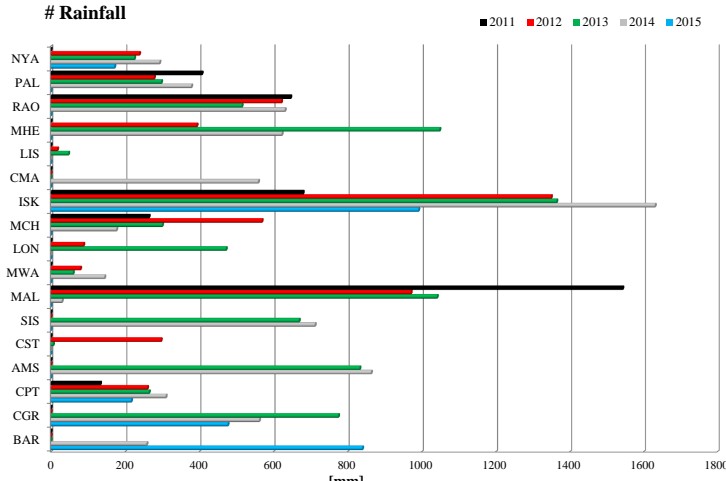

**Figure 3.** Precipitation amounts collected at all GMOS sites during 2011-2015

2013, and 2014 were respectively 5.8 $\mu g m^{-2} y^{-1}$, 6.5 $\mu g m^{-2} y^{-1}$, 4.2 $\mu g m^{-2} y^{-1}$, and 6.3 $\mu g m^{-2} y^{-1}$ at Rao site. This is more than two times higher than at Pallas during the same years (2.9 $\mu g m^{-2} y^{-1}$, 1.9 $\mu g m^{-2} y^{-1}$, 1.3 $\mu g m^{-2} y^{-1}$ and 2.3 $\mu g m^{-2} y^{-1}$), and since the precipitation amounts are also a factor of two higher at Rao site in comparison to Pallas, the Hg deposition results seem to be consistent with this increase in the south compared to the northern sites. These findings also

confirmed the results obtained by (Munthe et al., 2007) during an assessment on available Hg data in precipitation carried out from 1996 to 2002 at five Scandinavian EMEP monitoring stations, and among them also at Rao and Pallas GMOS sites. (Munthe et al., 2007) highlights, in fact, that the highest annual Hg wet deposition and yearly averaged THg concentrations in precipitation have been recorded at the southern Scandinavian coastal sites where the highest average annual deposition amounts also occurred. The annually based THg wet deposition flux ($\mu g m^{-2} y^{-1}$) calculated, conversely, at Mt. Changbai, Mt.

Walinguan and Mt. Ailao show no significant geographical trend with high variability and notable differences in concentrations among the sites during the same period. These stations are all remote sites in China, and considering the 2012, 2013, and 2014 period which is the most representative in terms of number of samples recorded, it is possible to see (Figure 2) that the averaged THg wet deposition fluxes ($n g m^{-2} d^{-1}$) in remote areas of China were not significantly higher than the values observed at the rest of the GMOS sites (i.e., ISK, MHE, RAO). At the sites located at lower latitude and Southern Hemisphere

the relationship between precipitation amount and deposition was not as evident as in the Northern hemisphere. At the Sisal monitoring station (SIS), a coastal site of the Tropical area located on the Yucatan peninsula (Gulf of Mexico), the 2013 annual wet THg deposition flux was 67.34 $\mu g m^{-2} y^{-1}$ and the average wet Hg deposition flux was 20.5 $n g m^{-2} d^{-1}$ whereas the rainfall amounts was 669.6 mm which is lower than the rainfall recorded at the remote southern sites, such as Amsterdam





Island (833.6 mm rainfall), and Cape Grim (775.6 mm rainfall) where the annual wet Hg deposition flux recorded were considerably lower at 1.95 and 3.1 $\mu gm^{-2}y^{-1}$, respectively, and the average wet Hg deposition flux as well at 7.2 and 10.6 $\mu gm^{-2}y^{-1}$, respectively (see Tables 2 and 3). The 2013 and 2014 annual wet deposition flux recorded at SIS are comparable or higher than those observed at most GMOS sites in the Northern and Southern Hemisphere (Tables 2 and 3). Because of

the Hg deposition at any given location is dependent upon both THg concentrations (which has a geographical component) in precipitation and precipitation amounts (Munthe et al., 2007), the results obtained across the sites located from the Tropical area to the Southern Hemisphere highlighted that in this case, the geographical component in terms of local meteorology and local emission sources, has had a higher influence on the THg results. During the sampling period SIS was typically influenced by air masses originated from Atlantic Ocean coming from east-south-east, but crossing the Caribbean Islands and/or Central/South

America with occasional air masses coming from east-north-east mostly during the winter period crossing the south of Florida and Caribbean Archipelago prior to arrive at the monitoring site (Sena et al., 2015; Sprovieri et al., 2016). Very few Hg deposition measurements have been performed at tropical latitudes (Hansen and Gay, 2013; Shanley et al., 2008);(Shanley et al., 2015). (Shanley et al., 2015) in a study over seven years (2005-2012) on Hg wet deposition at Puerto Rico (Caribbean Archipelago, US) highlighted that despite receiving prevailing unpolluted air off the Atlantic Ocean from northeasterly trade

winds, wet Hg deposition recorded at the site was about 30% higher than that observed in Florida and the Gulf Coast, which in turn, are the highest deposition areas in the U.S., and thus greater than at all other MDN sites. The wet Hg deposition map from the MDN, in fact, shows a general pattern of relatively low deposition over the western U.S. ($\sim 2-5\mu gm^{-2}y^{-1}$) and higher in the eastern U.S. (6-15 $\mu gm^{-2}y^{-1}$) due to increasing precipitation and location of important anthropogenic Hg sources. In addition, in the Eastern U.S. a north-south latitudinal gradient exists in wet Hg loading, with wet deposition reaching a

maximum in the SE U.S. over Florida (Prestbo and Gay, 2009; Selin, 2014). Despite its unpolluted, tropical setting, Puerto Rico seems to fit as a southern extension to a latitudinal gradient of increasing Hg deposition from north to south in the eastern U.S. (Shanley et al., 2015). The high wet Hg deposition at SIS can be directly linked to the meteo-climatic conditions and pressure systems typical of the tropics. The higher THg wet deposition observed at latitudes lower than south of Florida and or Mexico, such as Puerto Rico (27.9 $\mu gm^{-2}y^{-1}$) an unpolluted tropical site crossed often by air masses detected at SIS

prevalently in summer and fall and few in winter, also suggests that frequent high convective clouds in this subtropical region likely access the reservoir of oxidized Hg species in the upper free troposphere (Guentzel et al., 2001; Driscoll et al., 2013; Nair et al., 2013). (Shanley et al., 2015) found that the high Hg deposition was not correlated to GOM at ground level but to the maximum height of rain detected within clouds (obtained from the echo tops using the NOA-NEXRAD radar station) suggesting that droplets in high convective cloud tops scavenged GOM from above the mixing layer (Shanley et al. (2015)

and references therein). Numerous studies suggest in fact that the upper free troposphere holds a large pool of GOM that has been oxidized from the global Hg pool (Driscoll et al., 2013; Swartzendruber et al., 2006; Weiss-Penzias et al., 2009) and that frequent high convective clouds occurring in tropical regions, particularly closer to the Equator, scavenge GOM by precipitation being readily soluble (Lindberg et al., 2007; Selin and Jacob, 2008; Holmes et al., 2010). Closer to the equator, the Hadley cell structure indeed gives way to the Intertropical Convergence Zone (ICT), and the atmospheric circulation there may affect

upper-atmosphere Hg levels. The few measurements in the Northern-Hemisphere tropics, such as SIS, generally indicate lower



Hg fluxes than those measured at lower tropical latitude probably due to fewer convective rain events with clouds that reach the upper atmosphere (Shanley et al. (2015) and references therein). The higher annual wet Hg deposition observed at SIS compared to the other GMOS sites could be also due to a contribution of air masses crossing areas with discrete anthropogenic emission sources, particularly in late spring and summer, such as the metropolitan area of San Juan and/or minor industrial

plants in Fajardo and Antille Islands, and/or from air masses crossing, particularly in winter, several coal power plants and waste incinerations in the southern United States and southern Florida (Latysh and Wetherbee, 2007). In addition, also legal and/or illegal gold mining activities which are widespread (Veiga et al., 2006; Sprovieri et al., 2016) in the southern regions of the Yucatan peninsula (i.e., Nicaragua; Guatemala, etc.) could contribute to the Hg wet deposition at SIS. The southern sites, AMS, CPT, CGR, and Bariloche (BAR), Argentina are more remote compared to SIS. AMS is a very small island located

in the southern Indian Ocean where atmospheric Hg concentrations recorded during the same period were remarkably steady with annual median of $1.03\pm0.10\ ngm^{-3}$ and lower than those recorded at the Tropical sites (Angot et al., 2014);(Sprovieri et al., 2016) but slightly higher than annual averages and medians recorded at Cape Grim in 2013 (Slemr et al., 2014). Both AMS and Cape Grim for most of the time receive clean marine air masses (Slemr et al., 2014; Angot et al., 2014). Previous studies (Mason and Sheu, 2002; Sprovieri et al., 2003; Holmes et al., 2009; Sprovieri et al., 2010b, a) analyzed atmospheric

observations of GOM from Mediterranean, Pacific and Atlantic cruises in terms of Hg chemistry and deposition in the marine atmosphere, and suggested that elevated levels of halogen atoms, and in particular of Br in the marine boundary layer (MBL) are an important source of GOM from oxidation of GEM, that more readily deposited throughout sea-salt aerosols followed by aerosol deposition. GEM evasion from marine waters therefore, could represent a significant source of atmospheric Hg which contributes to depositional fluxes in marine regions (Mason and Sheu, 2002), such as Amsterdam Island, and Cape

Grim. In 2013, among the Southern sites, the highest annual and average THg wet deposition flux have been recorded at CPT ($5.2\ \mu gm^{-2}y^{-1}$ and $37.1\ ngm^{-2}d^{-1}$) which salso showed the lowest both deposition amount (264.9 mm) and the number of sampling days (Tables 2 and 3) compared to AMS (with annual wet deposition flux of $1.95\ \mu gm^{-2}y^{-1}$ and $7.2\ ngm^{-2}d^{-1}$, considering a rainfall of 833.2 mm) and CGR (with wet deposition flux of $3.1\ \mu gm^{-2}y^{-1}$ and $10.6\ ngm^{-2}d^{-1}$, considering a rainfall of 775.6 mm). These findings have not been observed at CPT in 2014 with the lowest annual wet deposition flux (0.57

$\mu gm^{-2}y^{-1}$) and comparable precipitation amounts and number of sampling days of the year before (see Tables 2 and 3).

CPT is situated on the southern tip of South Africa (Sprovieri et al., 2016; Brunke et al., 2016), and during the wetter season (May till October) normally precipitation increased due to the passage of cold fronts moving from West to East (Brunke et al., 2016). (Brunke et al., 2004) highlighted that CPT receives clean marine air most of the time whereas continental and polluted air masses are observed at the site more frequently during the winter period with air masses advected to the station from north

to north-western (Rautenbach and Smith, 2001; Brunke et al., 2004) region where the Gauteng and Mpumalanga provinces are located. These south African areas represent the major anthropogenic Hg sources with former mine dumps from gold mining and large coal-burning power stations (Dabrowski et al., 2008). Therefore, the highest annual average THg wet deposition flux observed at CPT in 2013 compared to the other southern sites which received more precipitation amounts than the CPT site seem to be prevalently influenced by regional/large scale emission sources during the sampling period. Measurements of

atmospheric Hg deposition in Bariloche (BAR), Argentina have been carried out for the first time from 2014 till 2015. BAR site



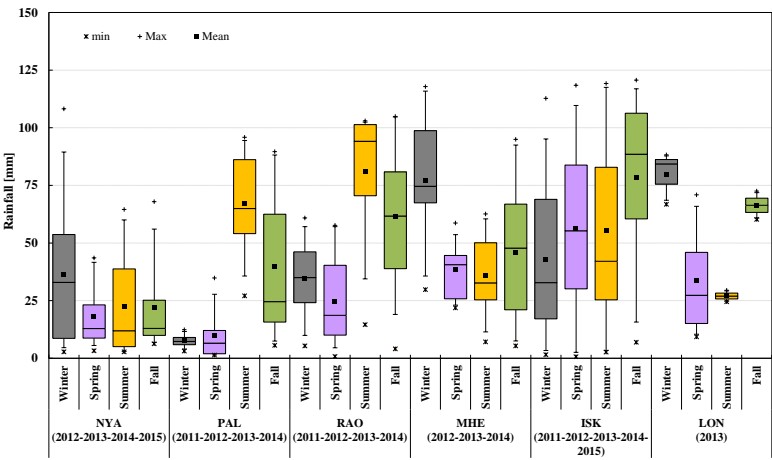

**Figure 4.** Seasonal distribution of rainfall amounts, at the European GMOS sites from 2011 to 2015

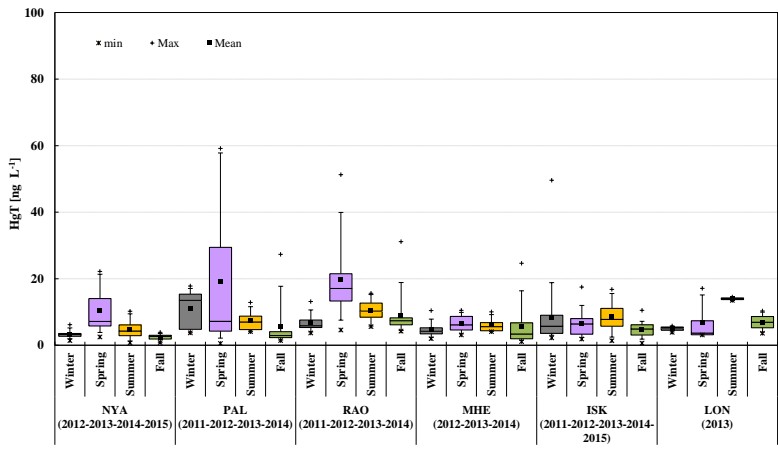

**Figure 5.** Seasonal distribution of volume-weighted THg concentration in precipitation at the European GMOS sites from 2011 to 2015





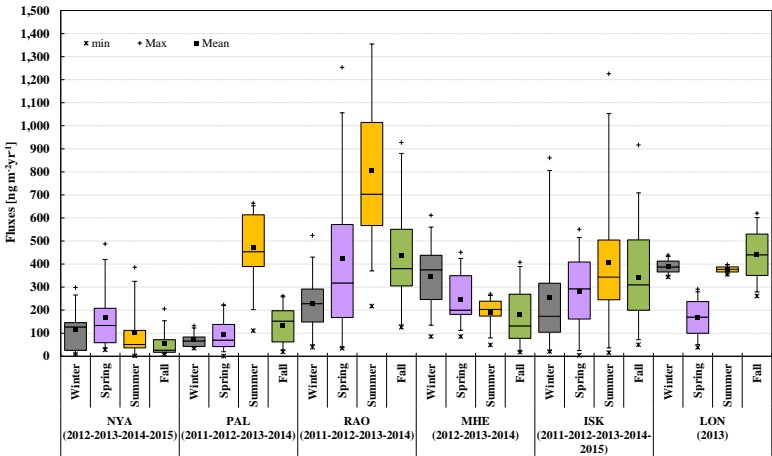

**Figure 6.** Seasonal distribution of THg wet deposition flux at the European GMOS sites from 2011 to 2015

has been established inside a well protected natural reserve in Northern Patagonia, on the shore of Gutierrez River at south-east of the Nahuel Huapi lake. GEM records at BAR station resemble background concentrations comparable to levels found in Antarctica and other remote locations of the South Hemisphere with annual mean GEM concentrations of $0.9 \pm 0.14\ ngm^{-3}$ (Diéguez et al., 2015; Sprovieri et al., 2016). The annual THg wet deposition flux calculated at BAR in 2014 was very low (0.1

$\mu gm^{-2}yr^{-1}$), however, it is necessary to point out that the number of samples carried out during the year was scarce (n = 91), therefore, the average wet deposition flux value (1.1 $ngm^{-2}d^{-1}$) obtained is less representative than that recorded in 2015 (3.0 $ngm^{-2}d^{-1}$) calculated over a number of sampling days of nearly 50% of the year. The 2015 THg wet deposition flux was 0.5 $\mu gm^{-2}yr^{-1}$ and an average wet deposition flux of 3.0 $ngm^{-2}d^{-1}$ which is lower than those recorded at the other southern GMOS sites with a comparable number of sampling days and, conversely, more close to the value observed in the Arctic, at

Ny Alesund station (4.2 $ngm^{-2}d^{-1}$).

## 4   Seasonal patterns and Influence of meteorological conditions on Hg wet deposition

### 4.1   European Stations

In this study, seasons are delineated according to the metereological definition. Since THg wet deposition flux depends on the total precipitation amount and the concentration of total Hg in that precipitation, the seasonal cycles of both these parameters

are shown along with the cycles of Hg wet deposition in Figures 4, 5, 6 and 7.





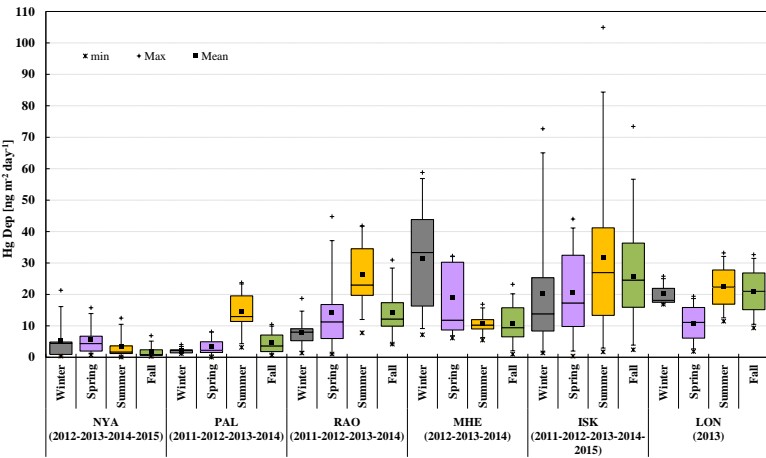

**Figure 7.** Seasonal distribution of THg wet deposition flux averaged on the number of sampling days, at the European GMOS sites from 2011 to 2015

Seasonal trends of THg in precipitation are clearly evident at all sites, with increased Hg concentrations and deposition observed during spring and summer months at most of them, implying a significant dependence on meteorological conditions throughout the years. The seasonal variability in Hg concentrations and Hg deposition has been reported in previous studies in North America (Hoyer et al., 1995; Landis and Keeler, 1997) and Europe (Iverfeldt, 1991; Munthe et al., 2007). The warm month

maximum in seasonal THg wet deposition is predominant at most European GMOS sites, except at Mace Head (MHE) and Longobucco (LON) where the maximum THg wet deposition occurs during the winter and the fall seasons, respectively. However, the patterns of THg concentrations and precipitation amounts reveal that at most of the sites, the seasonal THg wet deposition maximum corresponds to the maximum in precipitation amounts collected, except at Ny Alesund (NYA), Iskrba (ISK) and LON. Therefore, the dominant factor in determining the Hg wet deposition loading recorded at all the European

sites was generally related to the amounts of precipitation collected. Hg concentrations in rainfall at NYA peaked in spring, and decreased through the summer, in fall and winter seasons (Figure 5). Rainfall mean were fairly equally distributed in all seasons except the winter season. Thus, wet Hg loading was highest in spring, intermediate in winter and summer and lowest in fall (Figures 6 and 7). High levels of soluble species could in general be due to direct enhanced atmospheric oxidation of GEM to GOM, which occurs in regions with high concentrations of oxidants such as polar regions during springtime (where

AMDEs occur, such as NYA). At Pallas (PAL), Hg concentrations in rainfall increased through the winter, peaking in spring, and decreased through the summer and fall. Rainfall was not fairly equally distributed in all seasons but lowest values were





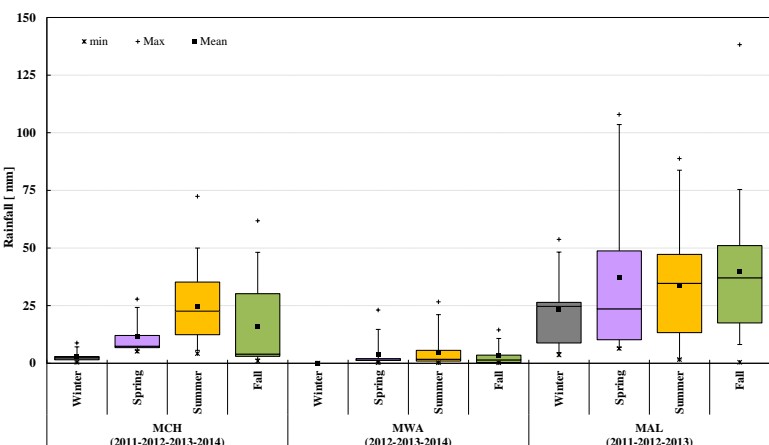

**Figure 8.** Seasonal distribution of rainfall amounts, at the three Chinese GMOS sites from 2011 to 2014

recorded during winter and spring and highest rainfall was observed in summer followed by a decreasing during the fall season. Thus, wet Hg loading was highest in summer, intermediate in fall, and lowest in winter and spring (Figures 6 and 7).

Similar rainfall behavior was observed at RAO, where Hg concentrations in rainfall peaked in spring, and decreased in fall and winter through the summer season. Therefore, wet Hg loading was highest in summer and the lowest in winter with intermediate values in spring and fall. At MHE, Hg concentrations in rainfall increased through the winter, peaked in spring, and decreased through the summer and fall seasons. Rainfall mean was fairly equally distributed in all seasons except the winter season. Thus, wet Hg loading was highest in winter, intermediate in spring and summer, and the lowest in fall (Figures 6 and 7). At ISK, Hg concentrations in rainfall increased from the winter, peaked in summer through spring, and decreased in fall. Rainfall mean was fairly equally distributed in spring and summer seasons except the winter season which shows the lowest rainfall whereas they peaked in fall season. Thus, wet Hg loading increased from the winter, peaked in summer through spring, and decreased in fall, following the same behavior of Hg concentrations in rainfall. (Figure 4). LON shows highest seasonal THg wet deposition in autumn and the lowest during spring. In this latter case, it is necessary to point out that these results are related to one year (2013) in contrast to the other sites in which all precipitation samples were grouped and analyzed season by season for a period of three to five years. Among the European sites the highest THg wet deposition have been recorded at the remote RAO and PAL stations during the more photochemically active summer months, whereas lower amounts were found in deposited in the colder months. In addition, rainfall amount during summer seems to be identified as the overriding factor controlling wet Hg loading at these sites. The lowest concentrations and total wet deposition were seen in winter months at most of sites. The seasonal pattern in the atmospheric Hg, with highest precipitation concentrations and wet deposition typically seen




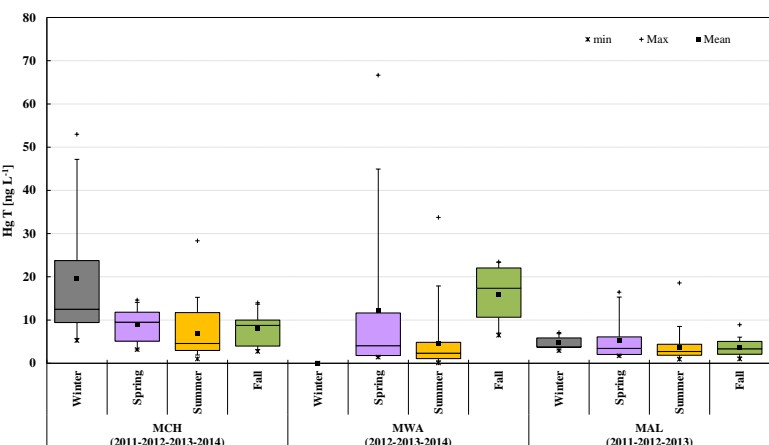

**Figure 9.** Seasonal distribution of volume-weighted THg concentration in precipitation at the three Chinese GMOS sites from 2011 to 2014

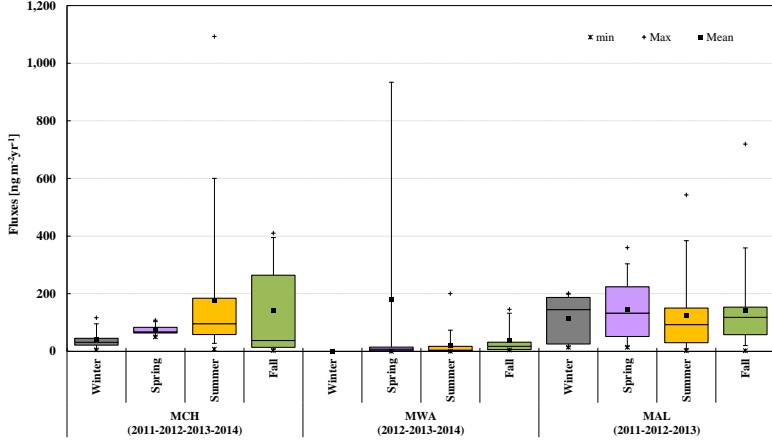

**Figure 10.** Seasonal distribution of THg wet deposition flux at the three Chinese GMOS sites from 2011 to 2014





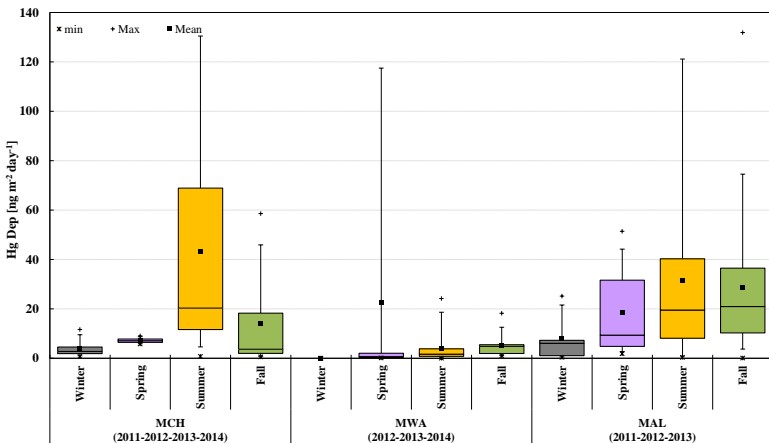

**Figure 11.** Seasonal distribution of THg wet deposition flux averaged on the number of sampling days, at the three Chinese GMOS sites from 2011 to 2014

in summer and lowest concentrations and wet deposition in winter, was believed partly to be the result of increased convection and mixing during the warmer summer months which can increase the ability of the air to transport Hg over longer distances, leading to greater precipitation amounts that remove Hg from the atmosphere. This may also indicate the role of precipitation type in the amount of Hg wet deposition, as rain may have a greater capacity to scavenge and hold different forms of Hg than

5  snow. Higher Hg deposition, typically observed during the warmer months, was likely the result of a mix of meteorological, source emission, and atmospheric chemistry influences. For example, it is widely known that the concentrations of oxidants such as ozone, OH radicals, and acids that oxidize GEM to GOM are higher during warmer months and would lead to elevated concentrations of oxidized species (Schroeder and Munthe, 1998; Lin and Pehkonen, 1999). Scavenging of soluble oxidized Hg species has also been considered to be more efficient in summertime precipitation events than in winter due to differences

10  in the cloud microphysical processing between rain and frozen precipitation (Hoyer et al., 1995).

## 4.2  Chinese Stations

China has been regarded as one of the largest atmospheric Hg emission sources region in the world (Streets et al., 2005; Wu et al., 2006). However, limited monitoring sites and data are available to understand Hg deposition patterns in China. Few previous measurements of THg deposition in China have been conducted in remote areas like Mt. Fanjing (Xiao et al., 1998),

15  Mt. Leigong (Fu et al., 2010), Wujiang River basin (Guo et al., 2008), and Mt. Gongga (Fu et al., 2008, 2010) in southwestern China, as well as at Mt. Changbai (Wan et al., 2009) in northeastern China. In order to evaluate the spatial and temporal





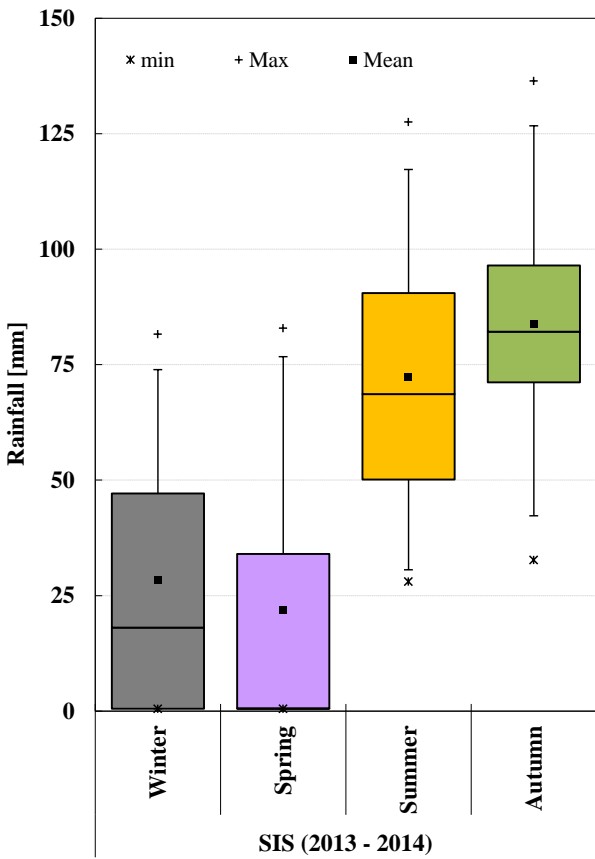

**Figure 12.** Seasonal distribution of rainfall amounts, at the tropical GMOS site (Sisal, Mexico) in 2013 and 2014

distribution of THg at the three GMOS Asian stations, all measurements performed from 2011 to 2014 at Mt. Changbai (MCH), Mt. Walinguan (MWA), and Mt. Ailao (MAL) were grouped by season and by site (Figures 8, 9, 10 and 11). Seasonal variations of THg in precipitation were observed at the three Chinese sites (Figure 9). The results obtained during the sampling period were similar to the seasonal variations of THg in precipitation in other Chinese regions, such as Wujiang River Basin, Guizhou,

5 China, but in contrast to the observations in North America (Landis et al., 2002), Adirondacks (Choi et al., 2008) and Great





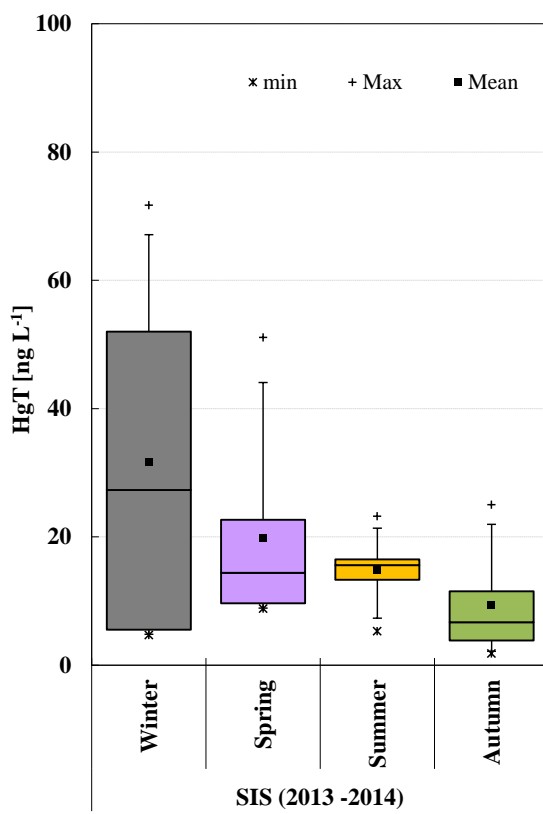

**Figure 13.** Seasonal distribution of volume-weighted THg concentration in precipitation, at the tropical GMOS site (Sisal, Mexico) in 2013 and 2014

Lakes region (Hall et al., 2005), which found increased THg concentration during summer months (Prestbo and Gay, 2009). Geographic differences in Hg wet deposition worldwide may be explained in part by the proximity to atmospheric sources and regional difference in anthropogenic emission sources. Atmospheric Hg species, in particular, GEM and PBM have been found to be substantially increased over recent years in both remote and urban areas of China, especially in central and eastern China,



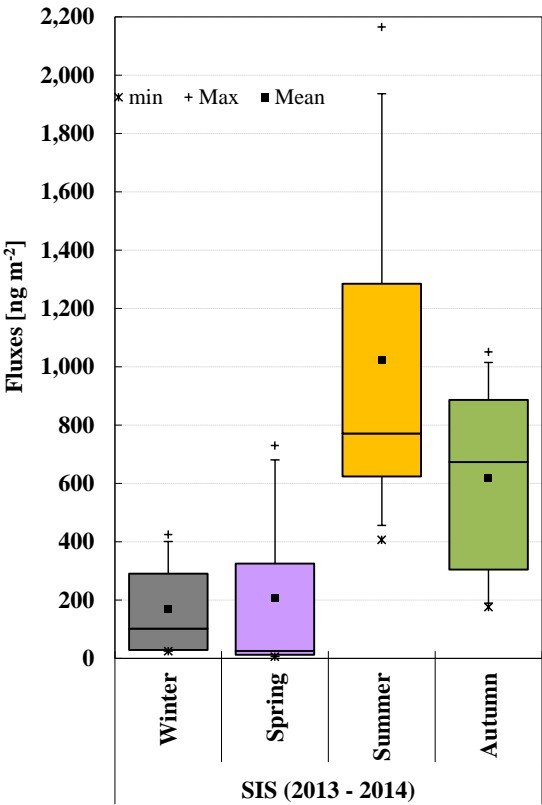

**Figure 14.** Seasonal distribution of THg wet deposition flux, at the tropical GMOS site (Sisal, Mexico) in 2013 and 2014

compared to those observed in North America and Europe which reported opposite long-term trends (Fu et al., 2015).The increasing trend in China is possibly caused by the increase in anthropogenic Hg emissions in the past decade, and indicates that the influence of regional emissions on Hg levels in China exceed global emission influence ((Lindberg et al., 2007) and references therein). The seasonal variation of weighted THg concentration observed in precipitation with highest value in winter and lowest in summer (see Figures 8, 9, 10 and 11), could be attributed in a first instance, to lower rainy amounts collected





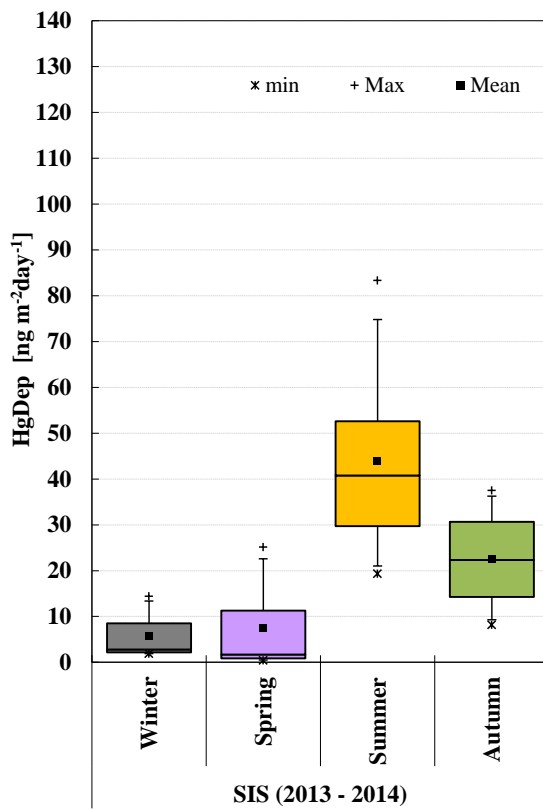

**Figure 15.** Seasonal distribution of THg wet deposition flux averaged on the number of sampling days, at the tropical GMOS site (Sisal, Mexico) in 2013 and 2014

in winter. The results obtained at the three Chinese sites show in fact that the THg concentrations varied with rain amount. In particular, at MCH, THg concentrations slightly increased in autumn, peaked during the winter season, and decreased during spring and summer when the lowest values were recorded. The reverse trend has been observed in precipitation amount through the seasons. Average THg wet deposition trend ($ng m^{-2} d^{-1}$) is comparable with that of the precipitation amount, with values





of THg flux increased from winter, through spring, and peaked in summer. Ruling out the winter season at MWA during which very few rainy samples have been collected, thus not representative for the present discussion, weighted THg concentrations peaked in fall and decreased during spring with lowest values in summer period. Therefore, wet Hg loading was highest in spring, intermediate in fall and lowest in summer. The positive or negative correlation between THg concentrations and the

precipitation amount has not been obviously observed at MAL where the rainy samples show a fairly variability during all seasons with lowest average rainfall in winter and the highest in fall, whereas THg concentrations showed high values in winter and lowest in fall, and wet Hg loading was highest in summer, intermediate in fall and spring and the lowest values were recorded in winter. (Fu et al., 2015) highlight significant positive correlations between rainwater THg concentrations and PBM and GOM concentrations, resulting in positive correlations between wet deposition fluxes and PBM and GOM concentrations.

This has been explained by the authors with the washout process of PBM and GOM during rain events which could contribute to enhance Hg wet deposition in China, particularly in urban areas where PBM and GOM concentrations are much higher. In remote areas of China, however, washout of elevated atmospheric PBM does not seem to drive a notable increase in Hg wet deposition flux, probably due to the low washout rate of PBM during rain events at high altitude monitoring sites, such as MAL and MWA where low-level clouds reduced the contribution of Hg washout (Lee et al., 2001; Seigneur et al., 2004).

(Guo et al., 2008) in a previous study in Guizhou on Hg in precipitation also pointed out that maximum THg concentrations in rainy samples during winter may be related to coal burning in domestic activities. Similar conclusions have also been reported in a study performed by Wang et al. (2012) at three Chinese sites (urban, residential and near-remote sites) in Chongqing province from 2010 to 2011, where they also found a high correlation between THg and particulate Hg (PBM) concentrations, suggesting that THg concentration in precipitation may be influenced by the PBM concentration. Additionally, comparable

seasonal behavior of Hg concentrations in precipitation with our results have been also observed, but with annual mean THg concentrations (ngL-1) significantly higher than those observed at MCH, MWA, and MAL sites which are located in remote Chinese areas. The seasonal pattern in deposition flux observed at the remote MCH, MAL, and MWA are comparable with those observed at remote sites of Europe and North America (Choi et al., 2008; Mason et al., 2000; Keeler et al., 2005; Sanei et al., 2010; Lombard et al., 2011), with maximum values during warmer months (Figures 8, 9, 10 and 11). It was suggested

by (Keeler et al., 2005) and (Mason et al., 2000) that this annual maximum was mainly due to more effective scavenging by rain in summer than by snow in the cold season (Sorensen et al., 1994; Mason et al., 2000; Keeler et al., 2005; Selin and Jacob, 2008).Mercury is not incorporated into cold cloud precipitation as efficiently as in warm cloud precipitation (Landis et al., 2002). Other explanations for this observation have been addressed by the authors including a greater availability of soluble Hg due to convective transport in summer events (Guentzel et al., 2001; Keeler et al., 2005), and a summer increase in

Hg-containing soil derived particles in the atmosphere (Sorensen et al., 1994).

### 4.2.1   Tropical Station: SISAL, Mexico

Hg deposition measurements are rare in tropical latitudes, with very few scientific publications in the past decade (Shanley et al. (2015) and references therein). The tropics are a particularly important region regarding global atmospheric chemistry. Due to intense ultraviolet radiation and high water vapor concentrations, high OH concentrations oxidize inorganic and organic gases,





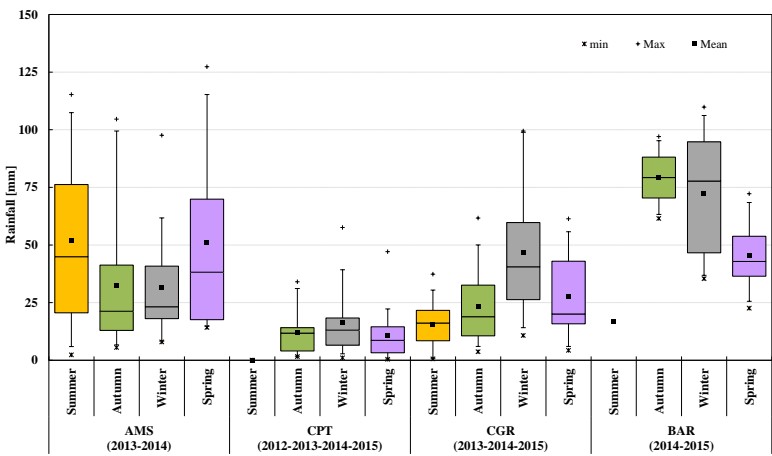

**Figure 16.** Seasonal distribution of rainfall amounts, at the four GMOS sites in the Southern Hemisphere from 2012 to 2015

and induce an efficient removal from the atmosphere of the oxidized products (Shanley et al. (2015) and references therein). Strong convective events in the tropical regions leads to huge volumes of air being drawn out of the sub-cloud layer with the resultant chemical composition of the precipitation coming from the capture of gases and small particles by the liquid phases of cloud and rain. Hg deposition measurements started in Mexico at Celestùn station (CST) in 2012 (see Table 1), but after a short

time period of sampling, the monitoring station changed the location with SIS, therefore, we refer the discussion to the SIS data related to both 2013 and 2014 years during which sufficient precipitation samples have been recorded. Despite receiving unpolluted air off the Atlantic Ocean from northeasterly and southeasterly trade winds, during most of the years (Sena et al., 2015), the site recorded higher wet Hg deposition fluxes during summer and fall compared to those observed during the other seasons. The SIS high Hg deposition rates, comparable to other sites in the Northern Hemisphere, such as the Chinese sites

(i.e., MWA) or European sites (i.e., ISK) that sometimes are also impacted by anthropogenic emissions, are driven in part by high rainfall events more intense during summer and fall, and less during winter and spring period. The high wet Hg deposition flux at this site suggests that other tropical areas may be hotspots for Hg deposition as well. A number of studies have suggested that this could be due to higher precipitation and the scavenging ratios from the global pool in the sub-tropical free troposphere where high concentrations of oxidized Hg species exist (Guentzel et al., 2001; Seigneur et al., 2004; Selin and Jacob, 2008).

These findings were also highlighted in previous studies in south of Florida and the Gulf of Mexico coastal areas confirming that local and regional Hg emissions play only a minor role on wet Hg deposition (Guentzel et al., 2001; Sillman et al., 2013) suggesting that the primary source of scavenged oxidized Hg could be the global pool.

Weather patterns in SIS exhibit a seasonality annual rainfall, with highest rainfall from June/July through October/November.




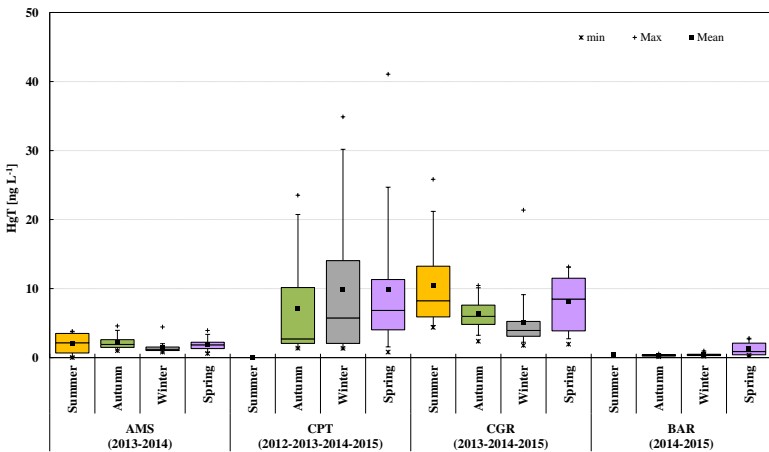

**Figure 17.** Seasonal distribution of volume-weighted THg concentration in precipitation, at the four GMOS sites in the Southern Hemisphere from 2012 to 2015

Summer tropical waves and systems characterized by deep convection and low pressure produced greater rainfall. During summer and fall, the site indeed receives rainfall from deep convection associated with tropical waves embedded in the prevailing easterly airflow. THg concentrations were higher in low volume samples. With larger storms Hg concentrations were diluted, this means that rainout of Hg was maximum (the decreasing of Hg concentrations with the increasing of the rainfall depth).

Weighted THg concentrations in rainfall ($ngL^{-1}$) increased from the fall, peaked in winter, and decreased through the spring and summer. On average terms THg in wet deposition was highest in summer, intermediate in fall, and lowest in spring and winter (Figures 12, 13, 14 and 15). The higher summer Hg deposition flux is not driven by higher Hg concentrations in rainfall since the highest Hg concentrations in rain samples occurred in winter (Figures 12, 13, 14 and 15). Different mechanisms leading to enhanced Hg concentrations in rain during the winter including greater anthropogenic emissions are probably as-

sociated with higher use of fossil flues in power plants during the cold season. As reported in Section 3 relating to the annual wet deposition patterns, the THg wet deposition observed at SIS could also be influenced by air masses crossing particularly in winter the southern Unite States and southern Florida where several coal power plants and waste incinerations (Latysh and Wetherbee, 2007) are located. The high wet deposition of Hg during the rainy seasons (May/June to October/November), in contrast, could be due to more efficient scavenging processes of reactive gaseous mercury from the free troposphere by tall

convective thunderstorms, and the concentration of GOM by the sea breeze effect, where the diurnal alternation of onshore and offshore winds can lead to a buildup of pollutants in the air mass. Greater information on Hg deposition and cycling is needed



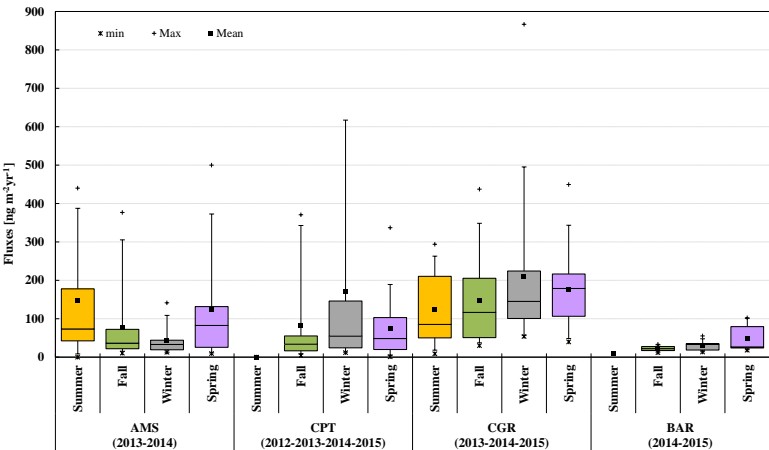

**Figure 18.** Seasonal distribution of THg wet deposition flux, at the four GMOS sites in the Southern Hemisphere from 2012 to 2015

in tropical regions, where populations are more likely to be exposed to Hg through fish consumption and artisanal gold mining activity.

### 4.2.2 Southern Hemisphere Stations

In remote areas far from any local sources, atmospheric deposition has been recognized as the main source of Hg to the ocean
(Lindberg et al., 2007; Pirrone et al., 2008). Hg can then be reemitted back to the atmosphere via gas exchange, and modeling studies suggest that reemission from oceans is a major contributor to atmospheric concentrations of GEM, particularly in the Southern Hemisphere where oceans were shown to contribute more than half of the surface atmospheric concentration ((Strode et al., 2007) and references therein). In the Southern Hemisphere we considered the four monitoring sites, Amsterdam Island (AMS), southern Indian Ocean, CPT, South Africa, Cape Grim (CGR), Australia, and Bariloche (BAR), Argentina which recorded a representative number of samples over the 2012-2015 period. Figures 16, 17, 18 and 19 show the box plots related to rainfall, THg concentrations in precipitation as well as wet deposition flux of Hg recorded at the four southern sites. An NSA-171 (Eigenbrodt) collector was set up at AMS at the beginning of the 2013. The GMOS site experiences a mild oceanic climate with monthly median air temperature ranged from 11 °C in austral winter to 17 °C in austral summer and frequent presence of clouds (Sciare et al., 2009). In 2013 and 2014 AMS displays the highest precipitation amounts collected during the warmer seasons (spring and summer) (Fig. 16, 17). Also the THg wet deposition flux patterns follow the same trend observed for the rainfall highlighting that the main factor driving the flux seems to be the amount of rain collected (Fig. 18). The THg fluxes





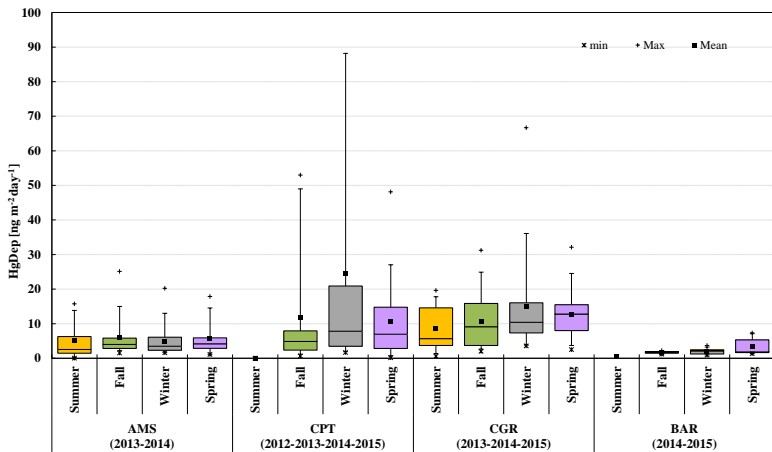

**Figure 19.** Seasonal distribution of THg wet deposition flux averaged on the number of sampling days, at the four GMOS sites in the Southern Hemisphere from 2012 to 2015

pattern seems to be in agreement with the results of atmospheric Hg speciation measurements carried out during the same period at AMS, and in particular with the GOM seasonal pattern observed since January 2012 by (Angot et al., 2014) that highlighted a higher frequency of GOM events between December and March (summer). However, additional and integrated measurements in ambient air and rainwater samples to improve our understanding of deposition processes and oxidation mechanisms should

be addressed. The variation of Hg concentrations in precipitation and Hg wet deposition fluxes driven by the precipitation amounts collected at AMS occurred also at CPT where, apart the dry summer season, Hg concentrations in precipitation, Hg wet deposition fluxes as well as the precipitation amounts, followed the same trend during the rainy season (May till October), with a maximum in wintertime for all the parameters recorded. CPT experiences a Mediterranean-type climate that is characterized by rather dry summers comprising moderate temperatures. The austral autumn to spring season normally experience increased

precipitation due to the passage of cold fronts moving from West to East, therefore, CPT generally receives clean marine air from the Atlantic Ocean whereas continental and polluted air masses are observed at the site more frequently, mainly during the winter period (Brunke et al., 2004, 2016), due to the prevailing air masses from the north to northwestern sector (Rautenbach and Smith, 2001; Brunke et al., 2004). The highest THg concentrations and wet deposition fluxes recorded during the winter season could be due also to the contribution of polluted air masses crossing Cape Town metropolitan area before arriving at

the stations. However, in a previous study on GEM concentrations and THg in precipitation carried out over a period of seven years (2007-2013) by (Brunke et al., 2016) highlighted that GEM, THg, CO and 222Rn levels within the urban-marine events observed at CPT did not substantially differ from those seen in the marine rain episodes, concluding that no significant local



anthropogenic influences were detected on THg concentrations. Conversely, a significant positive correlation was found CPT between GEM and THg concentrations, and with the Southern Oscillation Index (SOI), suggesting that both GEM and THg concentrations are primarily influenced by large scale meteorology which in turn controls Hg emission sources in terms, for example, of enhanced sea surface temperature that could increase large scale droughts leading to a raised biomass burning

(Brunke et al., 2016).

Measurements of atmospheric Hg deposition in Australia have never been reported before (Jardine and Bunn, 2010). From 2013 till 2015, at Cape Grim GAW Station (CGR), located on the north-western coast of Tasmania, Australia, highest value in rainfall have been observed during winter an lowest in summer, whereas Hg concentrations peaked in summer and dropped to lowest values in winter (see Fig. 16, 17, 18 and 19). The trend of Hg wet deposition fluxes conversely seems to be driven by

the precipitation amounts even if a small seasonal variability of Hg loading was displayed. Indeed, an increase in precipitation volume results in an increase of the Hg deposition flux. This is accompanied by a decrease in Hg concentrations in rain, probably due to the dilution of the washout loading (Prestbo and Gay, 2009). This means that any changes in meteorological conditions, especially precipitation, complicate the interpretation of GMOS observations at different latitude and might mask any trends due to change in Hg emissions. At BAR the highest precipitation amounts in 2014 and 2015 were collected during

the fall and winter seasons and decreased in spring when the highest THg concentrations occurred (see Fig. 16, 17, 18 and 19). Therefore, the seasonal THg wet deposition peaked in spring and decreased during the cold seasons. It is necessary to point out, however, that in 2014 at BAR no samples have been recorded in fall and summer as well as in 2015, during the same seasons the number of sampling days was very low particularly in summer. This means that further measurements and studies are needed to draw any conclusion and improve our understanding of deposition processes and oxidation mechanisms in this

region. There are very few previous observations of Hg wet deposition in the Southern Hemisphere, and this makes difficult any comparison of data recorded during GMOS. The results observed at the four southern GMOS sites highlighted that the magnitude of wet deposition is affected by two main factors: amount of precipitation and the THg concentration in precipitation influenced by soluble Hg species (oxidized Hg) in the atmosphere. High levels of soluble species could in general be due to direct anthropogenic emissions of Hg oxidized species or by enhanced atmospheric oxidation of GEM to GOM, which occurs

in regions with high concentrations of oxidants such as southern locations (where more solar radiation occurs) or polar regions during springtime (where AMDEs occur).

## 5 Conclusions

Mercury deposition measurements are critical for constructing an accurate global Hg budget and to model the benefits or consequences of changes in Hg emissions, for example, as proscribed by the Minamata Convention. Early models of wet Hg

deposition had few measurements for calibration or validation, and tended to overestimate the influence of local emission sources. A synthesis of all available Hg measurements in precipitation from GMOS network is presented, including trends and seasonal cycles. These results provide a set of data for modeling applications to fully understand THg wet deposition patterns as well as the transformation and deposition mechanisms of atmospheric Hg. With broad geographic coverage including mostly



background and remote sites with few local or regional sources, GMOS's observation network gives important insights to evaluate future Hg trends on global scale. The results on THg wet deposition carried out in this study open the way for new avenues in future modeling studies as well as highlight the need of additional and integrated measurements in ambient air and rainwater samples to improve our understanding of deposition processes and oxidation mechanisms. These new observations

in fact, give scientists and modelers some insight into baseline concentrations of THg concentrations in precipitation and depositional fluxes especially in the tropical area, and in the Southern Hemisphere where wet deposition as well as atmospheric Hg species were not investigated before. Greater information on Hg deposition and cycling is obviously needed in these regions. Moving forward, in addition to continued monitoring GMOS sites, integration with other ground-based monitoring sites at strategic locations along with integrations with atmospheric Hg species and other key oxidants, identification of the

compounds making up GOM and PBM continue to be needed. Knowledge of these exact chemical species would also lead to improved understanding of the chemistry and wet and dry deposition processes of oxidized Hg specie in different air masses. Wet deposition measurements worldwide would assist modelers in constraining the atmospheric Hg budget on global scale, as would additional direct measurements of dry deposition across the GMOS network.

*Acknowledgements.* This work was funded by the FP7 (2010-2015) Global Mercury Observation System (GMOS) project. We thank all

15 GMOS external Partners for providing high quality-controlled wet deposition measurements as well as we would like to acknowledge and thank all the site operators for the GMOS global network. AD, OM, HA thank the French Polar Institute IPEV(Program 1028, GMOStral), the LEFE CNRS/INSU (program SAMOA) and the overwintering crew at Amsterdam Island. The CNR-IIA research staff thanks also F. Cofone, A. Servidio, and A. Rosselli for their technical support for the laboratory work carried out on the samples from AMS, LON, CMA, and Mexican Stations.





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
