# Peer review of "Five-year records of mercury wet deposition flux at GMOS sites in the Northern and Southern Hemispheres"

_Atmospheric Chemistry and Physics, 2016_

## Referee Comment (RC1) · Anonymous Referee #1 · 29 Aug 2016

General comment:

The manuscript presents description and analysis of measurements of Hg wet deposition from the newly established GMOS monitoring network. Both spatial trends and seasonal variation of Hg concentration in precipitation and wet deposition fluxes are analyzed in relation to diverse meteorological and climatic conditions. Peculiarities, of Hg deposition in different locations are discussed including sites located in Europe, Asia, in the Tropics and in the Southern Hemisphere. Various factors affecting Hg wet deposition are considered. The subject of the manuscript is relevant to the scope of the journal and the work makes up a new and original contribution. The manuscript will be suitable for publication after addressing the specific comments mentioned below.

[Figure]

Specific comments:

Title: Commonly, "total deposition flux" means sum of wet and dry deposition. There is no discussion about Hg dry deposition in the paper. So, probably, "Five-year records of wet mercury deposition flux . . ." would be more proper title for the paper.

Page 2, line 19. *". . .in both depositional flux and concentration with the highest values. . ."* Page 2, line 25. *". . .coastal sites receiving higher Hg concentrations and depositional Fluxes. . ."*

Concentrations in air or in precipitation are mentions here? Please, specify to avoid misleading.

Page 2, lines 27-29. *". . .gaseous evasion of Hg from marine waters is a significant global source of atmospheric Hg and may also contribute to elevated depositional fluxes in coastal regions. . ."*

Wet deposition is mostly comprised precipitation removal of highly soluble oxidized Hg. Significant Hg evasion from the ocean, which is in poorly soluble elemental form, does not necessarily mean elevated deposition. Oxidation of GEM to GOM is essential.

Page 2, line 34. *". . .the EMEP program . . . the GMOS"*

These acronyms which appear for the first time require explanation and references.

Page 2, line 33 – Page 3, line 1. *"Long-term Hg wet deposition measurements exist at many locations within the United States as part of the MDN or in Europe as part of the EMEP program; however, before the establishment of the global Hg network by the GMOS, long-term of ambient Hg concentrations and measurements of Hg wet deposition fluxes were lacking"*

The second part of the sentence contradicts the first part.

Page 6, Section 2.3. What were the criteria of data coverage for calculation of annual and seasonal mean values? As it follows from Tables 2-3 and Figs. 1-3 the annual

data are not available for all the stations for all years.

Figs. 1-3. The bars at the diagrams are very thin that makes difficult reading the figures.

Page 8, lines 9-12. *"Deposition of atmospheric Hg at any given location is influenced by factors such as. . ."*

Since Hg deposition mostly consists of scavenging oxidized Hg forms (GOM or PBM) the list of factors should also include the oxidizing capacity of the atmosphere. The oxidation chemistry can be a dominating factor for Hg deposition at least in remote regions.

Page 10, lines 3. Probably, the units of the average wet Hg deposition flux should be ng m-2 d-1 instead of ug m-2 y-2.

Page 11, line 21. *". . .showed the lowest both deposition amount (264.9 mm) and. . ."* Should it be read as "precipitation amount"?

Pages 27-28. The Conclusions are too general and does not contain any particular findings on spatial trends and seasonal variation, factors affecting Hg wet deposition etc.

There are also numerous typos throughout the text which need spell-checking.

---

## Referee Comment (RC2) · Anonymous Referee #2 · 19 Sep 2016

**Review of the manuscript: Five-years records of the total mercury deposition flux at GMOS sites in the northern and southern hemisphere**

The paper by Sprovieri et al. deals with Hg deposition fluxes in N and S Hemispheres at different sites as a part of GMOS project. The paper provides comprehensive data and discussion in relation to temporal and spatial variability of Hg deposition fluxes on a global scale, particularly in regions, such as S Hemisphere and tropical areas where atmospheric Hg species have not been investigated before. These data present valuable information about the current status of Hg cycling as a global pollutant and for future modelling, which could be interesting to a broader scientific community. Even more, the study represents an important input for future planning and management in relation to environmental changes. I suggest publication after moderate revision taking into account the following comments:

1. The paper deals only by wet deposition. What about dry deposition? The title should be changed then to "…Hg wet deposition fluxes…"
2. P6 - What is the uncertainty of Hg wet deposition flux calculations?
3. The data in the manuscript should be provided according to the precision of measurements.
4. Although the Hg source identification is not the subject of the present paper it would be interesting to make a comparison with the study performed by Sun et al. ES&T, 2014. There are several parts in the manuscript where sources of Hg are discussed, which could be further supported by stable isotope analysis especially in China and their relation to global emissions (P20).
5. The spell-checking is needed throughout the paper – several typos present.

---

## Referee Comment (RC3) · M. Cohen (Referee) · 28 Sep 2016

Review of MS No.: acp-2016-517: "Five-year records of Total Mercury Deposition flux at GMOS sites in the Northern and Southern Hemispheres", by F. Sprovieri et al.

(Review by Mark Cohen, Sept 27, 2016)

**General Comments**

The authors present a detailed description of 2011-2015 mercury deposition measurements made at selected sites in Europe, Russia, China, Mexico, and the Southern Hemisphere. Tropical and Southern Hemisphere data have been heretofore particularly scarce, and so it is welcome to see these data presented, alongside data from the Northern Hemisphere. It is a difficult task to explain spatiotemporal variations in mercury wet deposition observations, and the authors are to be commended for their thoughtful discussion of the various factors that may have contributed to site-to-site variability. While there are some areas for potential improvement that can be considered, as discussed below, this is an excellent paper and should certainly be published.

**Specific Comments**

● The title is misleading, suggesting that both dry and wet deposition is being reported. Perhaps the title could be reworded to be something like the following: "Five-year records of mercury wet deposition flux at GMOS sites in the Northern and Southern Hemispheres"

● Page 2, Lines 8-11. Could mention that dry deposition is often estimated via models using measurements of ambient concentrations of mercury and meteorological parameters.

● Page 2, Lines 29-30. What is the gradient in northern Europe?

● Page 3, Lines 7-8; and page 5, Lines 6-8. Why were these particular 17 sites chosen out of the 43 monitoring stations worldwide? Why were some sites excluded?

● Table 1. Several questions and suggestions to consider:

> (a) "Elev." – could give units (m-asl) in the table. The units are given later in the text, but for clarity, could be included in the table.

> (b) The Sampling frequency could be included, e.g., 2-weeks for some sites, etc.

> (c) What is the meaning of sites listed as "M/S" and "S/M"?

> (d) If the site is a member of a national/regional network, this network could be listed.

> (e) The years of data collection could be noted for each site (e.g., 2013-2014, etc.)

> (f) In my opinion, would be very helpful to show the sites on a global map, perhaps with insets with close-ups as needed for clarity (e.g., for Europe)

● Table 2, and associated text. It is not clear to me what "ndays" data mean, and why it would be less than ~365 days per year. At some points in the manuscript, it seems that it might be being implied that if there was no precipitation during a given period (e.g., page 6, lines 20-21), then that period would not be reported as being a day of sampling for that year? But I don't think that this is what you mean. My understanding of wet deposition samples is that the collector is in the field for a certain period (e.g., 2 weeks) and any precipitation that falls during this period is collected. So, in the usual case, if the site is operational, then the sampling generally occurs for the entire year, i.e., ~365 days. There may be some sampling periods where no precipitation is collected, but this is still a "sample" to be counted in the number of sampling days for that year. So, it would be helpful to clarify what is meant for each site, for each year, when the number of sampling days is less than 365. Was the site "closed" for the non-sampling days, i.e., the collector was not being operated? And if this is the case, and since there are seasonal patterns to precipitation and mercury wet deposition, it is not clear to me that normalizing the measured deposition by the number of sampling days is a reasonable approach. In other words, the periods when the sampler was "on" would not necessarily be representative of the "average". I'm not sure if it's really useful to present data for fragments of years, given the seasonality, and given that the dates of collection are not given. In my opinion, it might be best just to give the data for a site when an entire year of samples was collected (or at least *most* of the year).

● Page 5, Lines 14-15:  Additional description could be given in the text regarding the "bulk-modified" sites, e.g., at least a few sentences describing the sampling protocol at these sites. E.g., what does "bulk-modified" mean? Also, should be noted that bulk-collection sites collect some dry deposition.

● Page 5, Lines 28-30. As noted above in comments on Table 1, it would be helpful to give the sampling frequency of each site.

● Figure 1. Several comments/suggestions:

  (a) Figure is too small to read easily. One suggestion would be to switch the x/y axes, i.e., put the sites along the x-axis on the bottom, and the flux on the y-axis. And then, use the whole width of the page, so that the data can be more easily distinguished. Another suggestion might be to use symbols rather than bars.

  (b) In the text, you refer to European sites extensively, and it would be helpful if these were grouped in the Figure. I know that you ordered the sites by latitude. But, in my opinion, you refer so many times to the "European sites" and refer to trends, etc., that it is really inconvenient to have to filter out the Chinese sites, etc.

  (c) Could consider showing a scatter plot of deposition flux vs. latitude instead, or in addition.

● Page 7, Lines 4-5. Seems that there are 11 sites in the Northern Hemisphere, rather than 10? And you discuss the European sites and Chinese sites extensively, but not the Russian site LIS.

Why is this? Also, here you say 7 European sites, but Figures 4-7 for European sites show only 6 sites. Could mention at some point why is the CMA site in Italy is not included in Figures 4-7.

● Page 7, Lines 7-9. The trend is not that clear in Figure 1, partly because the Russian and Chinese sites are interspersed in the Figure with the European sites. As noted above, I understand that you've listed the sites according to latitude, but ultimately, I think might be clearer if you group by region first, given that the discussion is predominantly carried out by region. Also, the LON site does not seem to fit the European trend noted.

● Page 7, Line 10. … no north-south spatial trend has been observed.

● Page 8, Lines 9-12. Wet deposition of atmospheric Hg at any given location…

● Page 8, Lines 9-12.  Wet deposition also depends on the type of precipitation (e.g., snow vs. rain), and the height and thickness of the precipitating cloud layer in the atmosphere, and the degree of convection involved. These are included at several points later in the document, but when I read this at this point, it seemed like important factors were being left out.

● Figure 3. As noted at several points in the manuscript, the relative proportion of snow vs. rain (or frozen vs. liquid) precipitation can be an important factor in interpreting the wet deposition data. Are there any site-specific data on this could be shown in Figure 3, or in a different figure?

● Page 9, Lines 1-9. The idea that more wet Hg deposition occurs with more precipitation is mentioned here and at several other points in the document. I think it might be really useful to show Figure(s) that show the deposition flux as a function of precipitation amount. This might be easier to interpret/explain than constantly going back and forth between the separate flux and the precipitation plots.

● Page 11, Lines 26-34. Here, you present arguments that suggest that the relatively high Hg wet deposition at CPT is due at least in part to contributions from local and regional sources. But then on page 26, lines 15-17, you cite a study that purportedly concluded that no significant local anthropogenic influences were found in Hg concentrations at CPT. How can these conflicting situations be reconciled?

● Figures 4-5-6-7. Are the box-whisker plots showing statistics for the sample-by-sample distributions for each season? If so, then it would definitely be important to know the sampling frequency. Might be useful to state what the boxes mean (25%, 50%, 75%?), and what the whiskers mean (5%, 95%?)

● Figure 5. Seems like could reduce y-axis to 0-60 ng/lit to show data more clearly.

● Figures 4-5-6-7: Again, maybe could add a figure that shows flux as a function of precip… This might be very illuminating.

● Figures 4-5-6-7: Why is CMA not included as a European station?  I guess because not enough data?

● Figures 4-5-6-7: Would be useful somehow to show degree of solid vs. liquid precip, e.g., in Figure 4, if these data were available.

● Page 13, Line 5: Why were only 91 days sampled at the site? Was this because there was no precipitation, or was this because the site was simply not operated during that time?

● Page 15, Lines 3-11. Here, and in some other places in the document, it seems that you are just restating the information that can be clearly seen in the Figures. The manuscript is pretty long, and perhaps some efficiency could be obtained by omitting at least some of this type of reiteration?

● Page 15, Lines 16-17. As mentioned earlier in other contexts, it might be really helpful here to show a graph of flux vs. precipitation amount.

● Page 17, Line 6. What emissions are larger in the warmer months?

● Page 19, Lines 2-3. Perhaps too much to ask, but would it be possible to show maps of emissions in relation to the sites?

● Page 21, Lines 1-2. Sorry to be repetitive, but again, could maybe show a graph of flux vs. precipitation amount.

● Page 22, Lines 10-15. Not sure what you mean by "washout". Are you referring to below-cloud scavenging of PBM by falling precipitation? Perhaps you could explain a bit more about the phenomena that you are describing here.

● Page 26, Lines 1-3. Here, and at a few other points, you note patterns in relation to GOM or other measurements. Would a graphic be useful here to show the relationship, e.g., GOM vs. volume-weighted-mean concentration in precip?

● Page 27, Lines 30-31. You state that early models tended to overestimate the influence of local emissions sources. This may or may not have been true, for one or more models, but I feel you'd need to cite a lot of different papers really make this statement. To me, seems like an overly provocative statement, and one that is not really needed for the paper? The general idea that observations are critical for model evaluation is certainly valid, but I don't think you can (or need to) make this sweeping statement about "early models". Indeed, Sunderland et al (2016) have recently pointed out that "early models" may have significantly underestimated the influence of local emissions sources!

> Sunderland, E. M., C. T. Driscoll, J. K. Hammitt, P. Grandjean, J. S. Evans, J. D. Blum, C. Y. Chen, D. C. Evers, D. A. Jaffe, R. P. Mason, S. Goho and W. Jacobs (2016). Benefits of Regulating Hazardous Air Pollutants from Coal and Oil Fired Utilities in the United States. *Environmental Science & Technology* 50(5): 2117-2120.

● Page 27, Line 31. Is this really all available GMOS wet dep data, or just the data from selected sites for selected years? Also, are the GMOS wet dep data (and other data?) available? Perhaps this could be mentioned?

● Page 28, Line 1. Having data a "remote" sites with few local or regional sources is important, for sure, but having data a sites that are influenced by local and regional sources are also important for better understanding of Hg atmospheric fate and transport (and model evaluation), etc.

**Technical Corrections**

● Page 3, Line 1: … long-term measurements of ambient Hg concentrations and  Hg wet deposition fluxes were lacking…

● Table 2. There is a vertical line in the top of the table (see clip below, with red circle), that I think should be removed.

| | | 2011 | | | | 2012 | | | | |
|---|---|---|---|---|---|---|---|---|---|---|
| | | Annual Wet Dep. Flux $[\mu g\, m^{-2}\, yr^{-1}]$ | Rainfall [mm] | ndays [d] | Weighted HgT $[ng L^{-1}]$ | Aver. Wet Dep. Flux $[ng m^{-2} d^{-1}]$ | Annual Wet Dep. Flux $[\mu g\, m^{-2}\, yr^{-1}]$ | Rainfall [mm] | ndays [d] | Weighted HgT $[ng L^{-1}]$ | Aver. Wet Dep. Flux $[ng m^{-2} d^{-1}]$ |
| Northern Hemisphere | NYA | - | - | - | - | - | 0,9 | 238,6 | 350 | 3,8 | 2,6 |
| | PAL | 2,9 | 407,4 | 363 | 7,1 | 8,0 | 1,9 | 278,6 | 332 | 6,8 | 5,7 |
| | RAO | 5,8 | 646,6 | 364 | 8,9 | 15,8 | 6,5 | 621,8 | 366 | 10,4 | 17,8 |
| | MHE | - | - | - | - | - | 0,9 | 393,7 | 113 | 2,2 | 7,6 |
| | LIS | - | - | - | - | - | **0,2** | **17,4** | **18** | **9,7** | **9,4** |
| | CMA | - | - | - | - | - | - | - | - | - | - |
| | ISK | 5,1 | 680,2 | 224 | 7,5 | 22,7 | 8,4 | 1349,7 | 363 | 6,2 | 23,2 |
| | MCH | 2,8 | 264,6 | 119 | 10,6 | 23,6 | 4,8 | 569,4 | 228 | 8,4 | 21,1 |
| | LON | - | - | - | - | - | **0,3** | **88,2** | **19** | **3,9** | **18,1** |
| | MWA | - | - | - | - | - | 0,3 | 79,5 | 127 | 4,3 | 2,7 |
| | MAL | 4,3 | 1543,2 | 222 | 2,8 | 19,5 | 3,2 | 971,5 | 202 | 3,3 | 16,1 |
| Tropics | SIS | - | - | - | - | - | - | - | - | - | - |
| | CST | - | - | - | - | - | 2,4 | 297,1 | 155 | 8,1 | 15,5 |
| Southern Hemisphere | AMS | - | - | - | - | - | - | - | - | - | - |
| | CPT | 0,3 | 133,5 | 119 | 2,1 | 2,4 | 3,8 | 260,3 | 147 | 14,6 | 25,8 |
| | CGR | - | - | - | - | - | - | - | - | - | - |
| | BAR | - | - | - | - | - | - | - | - | - | - |

● Table 3. I think "uom" refers to "units of measurement", but maybe clearer just to put the units, or spell out "units of measurement. Better yet to include the units directly in the table.

● Page 6, Lines 31-32: …the number of the sampling days as well as the annual wet deposition flux and average THg wet deposition flux calculated for each year in the period 2011-2015.

● Page 6, Line 32: As noted above, it is really unclear how valid any of the partial-year data are, given that it is unclear if the missing data are from rainy or dry seasons, etc.

● Page 8, Line 17. ... during the 2011-2015 period are  reported in Figure 3

● Page 11, Lines 19-24.  Seem like sometimes you refer to sites using the 3-letter abbreviation, and sometimes you refer the sites using the full name of the site. Since the graphics all use the 3-letter abbreviation, maybe better to just use these in the text throughout. Could give the full name the first time it was mentioned, with the abbrev in parentheses, and then just use the abbreviation from then on?

● Page 13, Line 13. "meteorological" is misspelled.

● Page 20, Line 5. "rain" not "rainy"

● Page 22, Line 5. … The positive or negative correlation between THg concentrations and the precipitation amount has not been obviously observed at MAL where the rain  shows  seasonal variability,  with lowest average rainfall in winter and the highest in fall…

● Page 23, Line 18. … exhibit  seasonality in  rainfall, …

● Page 24, Line 10. "fuels" not "flues"

● Page 24, Line 12. "United States"

● Page 24, Line 12. "waste incinerators"

---

## Author Comment (AC1) · 15 Nov 2016

First of all, we thank the three reviewers for their effort and useful suggestions reported for the manuscript on mercury wet deposition flux performed at the GMOS sites distributed worldwide. We completed the revision of the manuscript according to comments provided by reviewers taking into account the important input and corrections they highlighted.

We appreciate very much their valuable comments for improving the readability and interpretation of the manuscript. We think that after this review, our manuscript has been now improved. Below we report point by point our detailed responses to the comments for each Reviewer. Thank you very much once more.

**Anonymous Referee #1**

1. Title: Commonly, "total deposition flux" means sum of wet and dry deposition. There is no discussion about Hg dry deposition in the paper. So, probably, "Five-year records of wet mercury deposition flux...." would be more proper title for the paper.

**Reply:** We thank the reviewer and the other two reviewers (that highlighted the same comment) for pointing out an inaccurate title of the manuscript. We agree with you, therefore, following your input we revised the title according to. Please, see the Title of the revised version of the manuscript.

 Page 2, line 19. "....in both depositional flux and concentration with the highest values..." Page 2, line 25. "...coastal sites receiving higher Hg concentrations and depositional Fluxes...." Concentrations in air or in precipitation are mentions here? Please, specify to avoid misleading.

**Reply:** We thank this reviewer for pointing out an inaccurate statement in the original text. We have made thorough revision in the revised version of the paper to eliminate and/or correct such an inaccurate statement. Please, see page 2, lines 20-23 and 27-31 in the revised version of the manuscript. Thank you.

3. Page 2, lines 27-29. ".....gaseous evasion of Hg from marine waters is a significant global source of atmospheric Hg and may also contribute to elevated depositional fluxes in coastal regions....." Wet deposition is mostly comprised precipitation removal of highly soluble oxidized Hg. Significant Hg evasion from the ocean, which is in poorly soluble elemental form, does not necessarily mean elevated deposition. Oxidation of GEM to GOM is essential.

**Reply:** Yes, that's right. We agree with you in respect to the oxidation processes of GEM to GOM which are essential within the wet deposition mechanisms to remove the highly soluble form of Hg from the atmosphere. Therefore, in order to make the sentence more clear and accurate, we revised the sentence. Please, see the revised version of the manuscript at page 2, lines 31-33. Thank you.

4. Page 2, line 34. "....the EMEP program ..... the GMOS"...These acronyms which appear for the first time require explanation and references.

**Reply:** Done, Thank you. We explained these acronyms in the text, and reported the references. Please, see now at page 3, lines 5 and 6.

5. Page 2, line 33 – Page 3, line 1. "Long-term Hg wet deposition measurements exist at many locations within the United States as part of the MDN or in Europe as part of the EMEP program; however, before the establishment of the global Hg network by the GMOS, long-term of ambient Hg concentrations and measurements of Hg wet deposition fluxes were lacking" The second part of the sentence contradicts the first part.

**Reply:** Yes, Thank you for your comment. We revised these sentences according to in order to make clear what they mean. Please, see the revised text at page 3, lines 3-13.

6. Page 6, Section 2.3. What were the criteria of data coverage for calculation of annual and seasonal mean values? As it follows from Tables 2-3 and Figs. 1-3 the annual data are not available for all the stations for all years.

Reply: We appreciate very much the comment of the reviewer regarding the criteria of data coverage for calculation of annual and seasonal mean values. Regarding the data, unfortunately, we did not gain a full coverage as well as samples collected with an homogenous time frequency. For that concerning data coverage, the GMOS wet deposition samples have been carried out with irregularity due to technical troubles with stuff and some problems in situ. Otherwise, since many GMOS stations provided wet deposition data recorded for the first time at their locations, we believe that, even if with a partial coverage, they could be helpful in making possible a global picture of the issue under study. To overcome the irregularity in time-sampling frequency we have done a normalization of both rainfall amounts and Hg fluxes in respect to the ideal time-sampling period that is equal to 15 days, as previously established in our GMOS Standard Operating Procedures. Concerning the Tables 2 and 3 and the Figures 1-3, they have been replaced and reorganized in order to accommodate all comments and suggestions made on data coverage criteria and related calculation as well as on precision of measurements data etc., made also by the other two reviewers. Please, see Tables 2 and 3 that are replaced with Table S1 and Table S2 reported within the supplementary material added to the revised version of the manuscript. The supplementary material added also includes additional two new tables (Table S3 and Table S4) reporting useful information about measurements and data calculation. Figure 1 has been replaced with a scatter plot as suggested by the reviewer n.3 and Figure 2 and Figure 3 have been merged in a "new" Figure 2 in order to make clear the relationship of wet deposition fluxes and the rainfall amounts at each site for each year, and we have also switched x/y axes according to the reviewer n.3 and enlarged the bars to make easier reading the Figure itself. Please, see the revised version of the manuscript along with the supplementary material included. Thank you.

7. Figs. 1-3. The bars at the diagrams are very thin that makes difficult reading the figures.

**Reply:** Regarding Figure 1-3, according to your comment, and in order to accommodate also the comment of the third referee (see the comment of Mark Cohen Number 9 on Figure 1 within the "Specific Comments" section, and several his suggestion reported earlier related to the need to

combine fluxes vs. precip. amounts), Figure 1 has been replaced with the new Figure 1 (scatter plot), and Figure 2 and Figure 3 have been merged in an unique Figure 2 in which we reported the rainfall vs Fluxes. In addition, as you suggested, in the new Figure 2, the bars were enlarged in order to make more clear and easy to read them. Please, see the new Figures 1 at page 7 and 2 at page 9 in the revised version of the manuscript. We hope that the revised Figures (1-3) will meet the comments and suggestion of both the reviewers. Thank you for your input.

8. Page 8, lines 9-12. "Deposition of atmospheric Hg at any given location is influenced by factors such as: ::" Since Hg deposition mostly consists of scavenging oxidized Hg forms (GOM or PBM) the list of factors should also include the oxidizing capacity of the atmosphere. The oxidation chemistry can be a dominating factor for Hg deposition at least in remote regions.

**Reply:** Yes, valid comment related to the oxidizing capacity of the atmosphere as another important factor. We revised these sentences including this important factor in the text. Please, see now at page 8, lines 14-22. Thank you.

9. Page 10, lines 3. Probably, the units of the average wet Hg deposition flux should be ng m-2 d-1 instead of ug m-2 y-2.

**Reply:** We thanks this reviewer for this important point. According also to the other two Referees regarding the precision of the measurements as well as the method firstly adopted to calculate the deposition fluxes, based on the exact sampling days at each site for each year, (Please, see also the comments n.2 & 3 of the second referee and related our replies as well as the comment number 6 in "specific comments" of the third referee and related our reply), we decided to revise all calculation taking into account the sampling frequency at each site for each year, which unfortunately was not constant across the sites. For this reason we decided to normalize all rainfall and fluxes data taking into account the two-weeks (15 days) as reference period based on the standard operating procedure (SOP) for THg in precipitation adopted within GMOS network. Therefore, the units that firstly referred to the number of days during the first analysis of the results, have been ruled out the text according to the revised calculation performed. Please, see the revised text of the manuscript. Thank you.

10. Page 11, line 21. "...showed the lowest both deposition amount (264.9 mm) and..." Should it be read as "precipitation amount"?

Reply: Yes, that's right. We corrected it, thank you. Please, see at page 11, lines 21-22.

11. Pages 27-28. The Conclusions are too general and does not contain any particular findings on spatial trends and seasonal variation, factors affecting Hg wet deposition etc.

**Reply:** The conclusions have been integrated and reorganized according to your suggestion and comment. Please, see the revised version of the manuscript at the section "Conclusion", pages 27 and 28. Thank you.

12. There are also numerous typos throughout the text which need spell-checking.

**Reply:** Yes, thank you very much for highlighting this. We revised the manuscript to correct typos throughout the text. Please, see the corrections done within the whole new revised version of the manuscript which is now improved. Thank you once more.

**Anonymous Referee # 2**

1. The paper deals only by wet deposition. What about dry deposition? The title should be changed then to "...Hg wet deposition fluxes..."

Reply: Yes, we revised the Title according to. Please, see the revised version of the manuscript.

2. P6 - What is the uncertainty of Hg wet deposition flux calculations?

**Reply:** As reported within the manuscript, because of different meteorological and climatologically conditions of the sites, the precipitation was not collected over an entire year at each station due to limited amount of precipitation samples occurring during specific periods (i.e., dry seasons). Therefore, all flux calculations reported herein used rain depth determined from the bottle catch and the uncertainty of Hg wet deposition flux calculations is therefore strictly linked to the precipitation volume of each sample. Weighed sample aliquots (50-100 mL) are pretreated following the standard procedure reported within the manuscript. Average blank values was determined for each analytical run and subtracted to determine sample Hg concentrations. Average analytical uncertainty for Hg in precipitation has been calculated with the relative standard deviation (RDS), where the RDS is the standard deviation of the three replicate analysis of each sample. Low precipitation volume samples (<1.5 ml) have very high uncertainly and a very low impact on the volume-weighted mean concentration, and were therefore not reported. The bi-weekly precipitation volume-weighted mean (VWM) concentration was determined using data only from samples considered valid. In addition, all data regarding rainfall amounts and Hg fluxes have been normalized in respect to the ideal time-sampling period that is equal to 15 days, as previously established in our GMOS Standard Operating Procedures. Both rainfall and Hg fluxes have been reported and discussed within the paper by the box and whisker plots, showing the variability of the available samples collected at each site and each season of observation. Furthermore, in Table S3 and S4 of the Supplemental Material, we also provided, on annual basis, the number of single collected sample with related basic statistics (min, Max, mean and St. Dev.). Thank you for your comment.

3. The data in the manuscript should be provided according to the precision of measurements.

**Reply:** Thank you for your comment. We think that this comment is strictly linked to the previous comment reported above (n. 2). We appreciate very much the reviewer for pointing out this important issue. Please, see our reply reported above as well as the revised version of the manuscript which now also includes a supplementary material document. The supplementary material added to the revised manuscript also includes additional two new tables (Table S3 and Table S4) reporting useful information and statistics about measurements and data calculation. Please, see the revised version of the manuscript and the supplementary material included. Thank you.

4. Although the Hg source identification is not the subject of the present paper it would be interesting to make a comparison with the study performed by Sun et al. ES&T, 2014. There are several parts in the manuscript where sources of Hg are discussed, which could be further supported by stable isotope analysis especially in China and their relation to global emissions (P20).

**Reply**: We appreciate very much the suggestion of the reviewer regarding the work performed by Sun et al. (2014), therefore, to accommodate the above comments, we focused our attention on the results and interesting study performed on Hg stable isotope signatures of coal deposits worldwide and historical coal combustion emission. At the same time, as the reviewer pointed out the identification of Hg source is not the subject of the present paper, since in this study, we limit our discussions to assess Hg wet deposition fluxes at some GMOS site (including regions where Hg measurements in air and deposition have not performed before, such as Tropics and Southern Hemisphere) and their seasonal and inter-annual variation reporting somewhere possible impact/effects on data recorded from mixing of different emission sources based on previous discussion and interpretation published in the literature at several GMOS sites part of them already established before the GMOS network and published historical series of Hg data in ambient air and precipitation. Therefore, according to the above discussion (and discussion in the text) and due to the lack of direct evidence and/or experiments on Hg isotope fractionation during atmospheric Hg transformation and deposition in this study to better identify Hg emission sources, we found difficult to start and include here a discussion on this topic. This also to avoid just include a and/or few sentences which in contrast should require more investigation and studies on this important and very interesting issue. Therefore, we thanks to this reviewer to highlight this issue and will take into account the suggestion for a future work on Hg precipitation data and the influence that the contributions from different sources may be better explained throughout Hg isotope fractionation factors and composition which should be updated whenever more data are available also for modeling applications. Thank you once more for your suggestion and comments.

5. The spell-checking is needed throughout the paper – several typos present.

**Reply:** Yes, Thank you. Following your input and the input also by the other two reviewers on this issue, we revised throughout the manuscript the several typos and corrected them according to. Please, see the revised manuscript.

**Referee # 3 (Mark Cohen)**

**Specific Comments**

1) The title is misleading, suggesting that both dry and wet deposition is being reported. Perhaps the title could be reworded to be something like the following: "Five-year records of mercury wet deposition flux at GMOS sites in the Northern and Southern Hemispheres"

**Reply**: Yes, thank you. We revised the Title according to your suggestion as well as the other two reviewers. Please, see the new Title of the manuscript. Thank you.

2) Page 2, Lines 8-11. Could mention that dry deposition is often estimated via models using measurements of ambient concentrations of mercury and meteorological parameters.

**Reply**: Yes, that's right. Following your suggestion, we integrated the sentences according to. Please see at page 2, lines 8-12.

3) Page 2, Lines 29-30. What is the gradient in northern Europe?

**Reply**: The north-south gradient in atmospheric mercury concentrations in northern Europe has been discussed in a manuscript published by Wangberg et al. (2001) (Atmospheric Environment Journal) which reports a summary of the results obtained in the framework of two joint MAMCS-MOE EU-funded projects, and in particular, the above paper reports the results from the MOE (Mercury Over Europe) project, in which has been discussed the north-south gradient for TGM with the highest values in the south considering measurements of TGM performed at six monitoring sites in North Europe, and confirming these findings reported in previous studies at the same sites (i.e., Smolke et al., 1999). In the manuscript Sprovieri et al. (2016) this has also been deeply discussed in the framework of the GMOS global network, thus considering several monitoring sites and confirming the gradient north-south, but extending the discussion from the Northern Hemisphere to the Southern Hemisphere, throughout the Tropical areas. Please, see Sprovieri et al. (2016) published in ACP journal. In this paper, in fact, there is a whole Section (Section 4.2) dedicated to the Northern – Southern hemispherical gradients in which we calculated the probability density functions (PDFs) of the 2013 and 2014 data and related histograms following the Scott rule, and, considering the mean (X) of the experimental measures for the northern (XN), southern (XS) and tropical (XT) groups and the confidence intervals evaluated from the Student t test among them, we observed a clear gradient of GEM concentrations from the Northern to the Southern Hemisphere (XN >XT>XS). The spatial gradient observed from northern to southern regions is also highlighted in two Figures (5 and 6), which also report the statistical monthly distribution of GEM values at all GMOS sites in the Northern and Southern hemispheres as well as in the tropical area. The gradients calculated are reported within the following table:

| Northern (XN), Southern (XS)
and Tropical (XT) | Mean X
(2013-2014) | Δ ng m -3 |
|---------------------------------------------------|-----------------------|----------------------|
| XN                                                | 1.54                  | 0.31                 |
| ХТ                                                | 1.24                  | 0.25                 |
| XS                                                | 0.99                  | 0.56                 |

For major details, please, see the paper Sprovieri et al. (2016), ACP. Thank you for your comment.

4) Page 3, Lines 7-8; and page 5, Lines 6-8. Why were these particular 17 sites chosen out of the 43 monitoring stations worldwide? Why were some sites excluded?

**Reply**: We thanks the reviewer for point out this important issue. The GMOS network to date consists of 43 monitoring sites globally distributed, and it has been established within the EU-founded project "GMOS" which start at the end of 2010. Several sites managed by the GMOS partners received founds to establish their own site during the development of the project, therefore, not all sites started together all Hg measurements in ambient air and precipitation, therefore, we chosen the most representative sites looking at their location (i.e., southern hemisphere, tropical

sites etc.) and their data coverage in order to have a number scientifically representative of results to discuss the data in different regions of the world. Obviously, this is a first tentative to discuss all together data from sites "globally" distributed taking into account that there are yet several difficulties linked to different meteorological conditions, emission sources and so on. Therefore, it well known for us that additional sites covering different regions and further investigations need to better understand the mercury fate and transport. GMOS network is ongoing and other additional sites are adding to the global network, therefore, we hope to collect in the next future more information and valid data to be discuss along with modeling applications for providing new insights in the mercury chemistry on regional and global scale. Thank you once more for your comment and suggestion.

- 5) Table 1. Several questions and suggestions to consider:
- (a) "Elev." could give units (m-asl) in the table. The units are given later in the text, but for clarity, could be included in the table.
- (b) The Sampling frequency could be included, e.g., 2-weeks for some sites, etc.
- (c) What is the meaning of sites listed as "M/S" and "S/M"?
- (d) If the site is a member of a national/regional network, this network could be listed.
- (e) The years of data collection could be noted for each site (e.g., 2013-2014, etc.)
- (f) In my opinion, would be very helpful to show the sites on a global map, perhaps with insets with close-ups as needed for clarity (e.g., for Europe)

**Reply**: We thanks the reviewer for point out several suggestions related to Table 1. We including them (point: a, c, d, and e) according to. Regarding point (f) we included a map named as Figure 1S within the supplementary material document added to the manuscript. Please, see the revised Table 1 in the revised version of the manuscript at page 4 of the revised version of the paper, as well as the supplementary material. Regarding the point (b) on the sampling frequency for each site, we included within the supplementary material two new tables (S3 and S4) in which we reported the exact number of the total yearly sampling days as well as the total number of single samples for each site. From these detailed information, it is possible to draw the sampling frequency related to each site for each year. Unfortunately the sampling frequency was not constant across the sites. For this reason we decided to normalize all rainfall and fluxes data taking into account the two-weeks (15 days) as reference period based on the standard operating procedure (SOP) for THg in precipitation adopted within GMOS network. Thank you very much for your input and suggestions.

6) Table 2, and associated text. It is not clear to me what "n days" data mean, and why it would be less than ~365 days per year. At some points in the manuscript, it seems that it might be being implied that if there was no precipitation during a given period (e.g., page 6, lines 20-21), then that period would not be reported as being a day of sampling for that year? But I don't think that this is what you mean. My understanding of wet deposition samples is that the collector is in the field for a certain period (e.g., 2 weeks) and any precipitation that falls during this period is collected. So, in the usual case, if the site is operational, then the sampling generally occurs for the entire year, i.e., ~365 days. There may be some sampling periods where no precipitation is collected, but this is still a "sample" to be counted in the

number of sampling days for that year. So, it would be helpful to clarify what is meant for each site, for each year, when the number of sampling days is less than 365. Was the site "closed" for the non-sampling days, i.e., the collector was not being operated? And if this is the case, and since there are seasonal patterns to precipitation and mercury wet deposition, it is not clear to me that normalizing the measured deposition by the number of sampling days is a reasonable approach. In other words, the periods when the sampler was "on" would not necessarily be representative of the "average". I'm not sure if it's really useful to present data for fragments of years, given the seasonality, and given that the dates of collection are not given. In my opinion, it might be best just to give the data for a site when an entire year of samples was collected (or at least most of the year).

**Reply**: We appreciate very much the reviewer for pointing out this important issue. We think that this comment is strictly linked to the previous comment reported above (Referee 1, comment n.6). Therefore, please, see also our related reply to the comment n.6) as well as our reply to your comment reported above (n.5). The "n days" means the number of days during which each station was able to collect wet deposition samples. Unfortunately, GMOS wet deposition samples have been carried out with irregularity due to technical troubles with stuff and some problems in situ mainly found at those stations that began their Hg monitoring within the GMOS project. Therefore, when we had less than 365 days per year it means that, even if the corresponding station was "open", the collector was not being operated. Otherwise, since many GMOS stations provided wet deposition data recorded for the first time at their locations, we believe that, even if with a partial coverage, they could be helpful in making possible a global picture of the issue under study. With this in mind, in this revised version of the paper we also tried to improve the presentation of data by normalizing them in respect of the ideal time-sampling period that is equal to 15 days, as previously established in our GMOS Standard Operating Procedures. In this way, the box and whisker plots, now report a consistent picture showing the variability of the wet deposition samples available on seasonal basis. Furthermore, in this revised version of the paper, we included additional information regarding the variability on annual basis of the wet deposition fluxes available at each GMOS site (See Tables S3 and S4 in the Supplemental Material).

7) Page 5, Lines 14-15: Additional description could be given in the text regarding the "bulk-modified" sites, e.g., at least a few sentences describing the sampling protocol at these sites. E.g., what does "bulk-modified" mean? Also, should be noted that bulk-collection sites collect some dry deposition.

**Reply**: We thanks the reviewer for point out this issue. We integrated this part of the manuscript following this comment (please, see the revised version of the manuscript at pages 4 and 5, lines 10 and 1- 4, respectively) including also the reference in which a detailed description of the bulk sampler used within the network is reported and its equivalence with the wet-only collectors. In addition, we replaced in Table 1 the "Bulk-modify" with the "IVL-Bulk" sampler which is the correct definition of the samplers that the sites reported in Table 1 used. Thank you for your comment.

8) Page 5, Lines 28-30. As noted above in comments on Table 1, it would be helpful to give the sampling frequency of each site.

**Reply**: Yes, thank you. Please, see our reply to your previous comment n.5 and the additional Tables reported within the supplementary material document added to the manuscript.

9) Figure 1. Several comments/suggestions:

(a) Figure is too small to read easily. One suggestion would be to switch the x/y axes, i.e., put the sites along the x-axis on the bottom, and the flux on the y-axis. And then, use the whole width of the page, so that the data can be more easily distinguished. Another suggestion might be to use symbols rather than bars.

(b) In the text, you refer to European sites extensively, and it would be helpful if these were grouped in the Figure. I know that you ordered the sites by latitude. But, in my opinion, you refer so many times to the "European sites" and refer to trends, etc., that it is really inconvenient to have to filter out the Chinese sites, etc.

(c) Could consider showing a scatter plot of deposition flux vs. latitude instead, or in addition.

**Reply**: We thanks the reviewer for detailed comments and suggestions. We considered at the end firstly the suggestion (c) replacing the old Figure 1 with a scatter plot of deposition flux vs latitude (please, see new Figure 1 at page 7); in addition, we also reported the deposition flux and the rainfall as an unique figure (see Figure 2 at page 9) in order to make more clear and simplify the interpretation/explain the results and avoid constantly going back and forth between the separate flux and the precipitation plots, as you also suggested within your comment reported later (n.16). Moreover, in order to improve and to make easier the Figure 2 reported in this part of the manuscript, we followed also your suggestion (a) switching the x/y axes, and enlarging the bars so that the data can be more easily distinguished. In addition, taking into account the comment n. 16 of the reviewer reported below and elsewhere, we re-organized all the Figure(s) related to both Section 3 and Section 4, including in each sub-section a Figure reporting the deposition flux as a function of precipitation amount according to. Thank you once more for your input and effort.

10) Page 7, Lines 4-5. Seems that there are 11 sites in the Northern Hemisphere, rather than 10? And you discuss the European sites and Chinese sites extensively, but not the Russian site LIS. Why is this? Also, here you say 7 European sites, but Figures 4-7 for European sites show only 6 sites. Could mention at some point why is the CMA site in Italy is not included in Figures 4-7.

**Reply**: We thanks to the reviewer for point out the number of sites in the Northern Hemisphere. There was a mistake in the text. In the revised manuscript we corrected the number of the sites at page 6, line 26, replacing 10 with 11 sites. We didn't discuss the results related to the Russian site because of the number of rainy samples was unfortunately inconsistent for any discussion or conclusion also from statistical point of view. Regarding CMA we didn't included within the seasonal variation graphics because for this site we have available few samples related to only two seasons and thus not covering a whole year (see Tables S1, S2, S3, S4). Therefore, we decided to rule out the site from the discussion of the re-organized Figures 4-7. The Figures 4-7, now became Figure 3, 4 and 5. Please, see the revised Figures at pages 13, 14 and 15, respectively, included within the revised manuscript. Thank you.

11) Page 7, Lines 7-9. The trend is not that clear in Figure 1, partly because the Russian and Chinese sites are interspersed in the Figure with the European sites. As noted above, I understand that you've listed the sites according to latitude, but ultimately, I think might be

clearer if you group by region first, given that the discussion is predominantly carried out by region. Also, the LON site does not seem to fit the European trend noted.

Reply: We thanks to the reviewer for this suggestion. However, as reported in our reply above, Figure 1 has been replaced with the new Figure 1 which consists in a scatter plot as suggested by this reviewer leaving the x/y axes as the original Figure 1 in order to make, in our opinion, more clear for the readers the general trend of the data observed according to the latitude. In addition, the re-organization of the Figures (firstly Fig.s 1-3, and now Figure 1 and Figure 2) made following your previous comments and the suggestion repeated later, could be from our point of view, a good compromise to better understand the discussion presented here and elsewhere not penalizing the general trend observed across the sites, particularly in the Northern Hemisphere. The Figures have been reorganized taking into account the precipitation amounts for each site and for each year in corresponding with the wet deposition fluxes as suggested. Please, see the new Figures 1 and 2 at pages 7 and 9, respectively. In addition, even according to your suggestion, in the Figure 2 we switched the x/y axes and enlarged the bars which are more clearly visible and in vertical position. Please, see our reply to your comment and suggestion n.9. We hope now that these changes could help to make more clear the discussion and interpretation of the results. Regarding LON site, we corrected this sentence in agreement with you. Please, see at page 8, lines 6-8 in the revised version of the manuscript. Thank you for your input and suggestions.

12) Page 7, Line 10. ... no north-south spatial trend has been observed.

**Reply**: Yes, that's right. We integrated the sentence according to. Please, see at page 7, lines 1-2, thank you.

13) Page 8, Lines 9-12. Wet deposition of atmospheric Hg at any given location...

Reply: Corrected, please, see the revised version of the manuscript at page 8, line 14. Thank you.

14) Page 8, Lines 9-12. Wet deposition also depends on the type of precipitation (e.g., snow vs. rain), and the height and thickness of the precipitating cloud layer in the atmosphere, and the degree of convection involved. These are included at several points later in the document, but when I read this at this point, it seemed like important factors were being left out.

**Reply**: Yes, that's right. We integrated these factors according to along with other important factors suggested by the Referee 1. Please, see the revised manuscript at page 8, lines 15-22. Thank you.

15) Figure 3. As noted at several points in the manuscript, the relative proportion of snow vs. rain (or frozen vs. liquid) precipitation can be an important factor in interpreting the wet deposition data. Are there any site-specific data on this could be shown in Figure 3, or in a different figure?

**Reply**: Yes, we agree with you. Unfortunately, we have no info/data from the sites on this. They didn't make this classification of each sample recorded. This also probably due to some logistic difficulties sometimes to reach the remote sites.

16) Page 9, Lines 1-9. The idea that more wet Hg deposition occurs with more precipitation is mentioned here and at several other points in the document. I think it might be really useful

to show Figure(s) that show the deposition flux as a function of precipitation amount. This might be easier to interpret/explain than constantly going back and forth between the separate flux and the precipitation plots.

**Reply**: We agree with your suggestion. We re-organized all the Figure(s) related to both Section 3 and Section 4, including in each sub-section a Figure reporting the deposition flux as a function of precipitation amount according to. Please, see also our reply to the comment n. 9 in which we listed the revisions made in both manuscript and Figures. Please, see the revised manuscript and the new Figures reported in the Sections 3 and 4. Thank you.

17) Page 11, Lines 26-34. Here, you present arguments that suggest that the relatively high Hg wet deposition at CPT is due at least in part to contributions from local and regional sources. But then on page 26, lines 15-17, you cite a study that purportedly concluded that no significant local anthropogenic influences were found in Hg concentrations at CPT. How can these conflicting situations be reconciled?

**Reply**: We thanks the reviewer for highlighting this important point. In this part of the text we reported some research and results obtained at CPT in the past by the authors cited in the paper, and in particular we referred our results of Hg wet deposition observed in 2013 to a study performed by Brunke et al. in 2014, and assuming as first instance that these high Hg concentrations could be due to influences from anthropogenic sources considering the origin of air masses prevailing at CPT during specific period of the year (Please, see the revised version of the text at page 10, lines 5-8). However, as pointed out at page 26 in the revised manuscript, lines 2-10, we also reported the results obtained in a more recent work by the same authors (Brunke et al., 2016) where, throughout additional measurements performed, they concluded that during the same period of the year the high Hg concentrations recorded at CPT are probably related to other processes occurring more than to significant local anthropogenic emission sources, and highlighting the positive correlation they found between GEM/THg concentrations, and the Southern Oscillation Index (SOI) giving such conclusion:... "suggesting that both GEM and THg concentrations are primarily influenced by large scale meteorology which in turn controls Hg emission sources in terms, for example, of enhanced sea surface temperature that could increase large scale droughts leading to a raised biomass burning"... In addition, we have made thorough revision in the paper to correct inaccurate words to make clear the sentences. Please, see the revised version of the manuscript related to this issue at both page 10 and page 26. Thank you once more.

18) Figures 4-5-6-7. Are the box-whisker plots showing statistics for the sample-by-sample distributions for each season? If so, then it would definitely be important to know the sampling frequency. Might be useful to state what the boxes mean (25%, 50%, 75%?), and what the whiskers mean (5%, 95%?)

**Reply**: We thanks the reviewer for highlighting this important point. Regarding the sampling frequency, please, see our reply to your comments n. 5, 6 and 8 and the related information reported within the new Tables (S3 and S4) in the supplementary material added to the manuscript from which is possible to understand the sampling frequency related to each site. In addition, following your suggestion, we reported what the boxes mean and the whiskers mean within the caption of the Figures. Figures 4-5-6-7 as well as the others following them included within Section 4, have been replaced/re-organized following your comment and suggestion reported elsewhere, therefore, the

new Figures have also a new numeration (Figures 4-5-6-7 are now 3, 4 and 5) and the other accordingly. Please, see the Figures and captions within Section 3 and Section 4. Thank you.

19) Figure 5. Seems like could reduce y-axis to 0-60 ng/lit to show data more clearly.

**Reply**: yes, we agree with you and we revised the old Figure 5 (now Figure 3) according to. Please, see the revised version of the manuscript. Thank you.

20) Figures 4-5-6-7: Again, maybe could add a figure that shows flux as a function of precip... This might be very illuminating.

**Reply**: Done. Please, see our reply to your comment n. 16 and suggestion reported elsewhere on this issue. Please, see the revised version of the Figures included within the manuscript. Thank you.

21) Figures -5-6-7: Why is CMA not included as a European station? I guess because not enough data?

**Reply**: Yes, that's right, the coverage related to this site is not enough consistent to drawn interesting conclusion and/or discussion. Please, see also our reply to your comment n. 10 on this issue. Thank you.

22) Figures 4-5-6-7: Would be useful somehow to show degree of solid vs. liquid precip, e.g., in Figure 4, if these data were available.

**Reply**: Unfortunately, we have no data and detailed information from the sites on this. Please, see also our reply to the above comment n.15.

23) Page 13, Line 5: Why were only 91 days sampled at the site? Was this because there was no precipitation, or was this because the site was simply not operated during that time?

**Reply**: Yes, the last one. The BAR site started the THg measurements with a delay in respect to most of the GMOS sites considered in the paper. Please, see also our reply to your comment n.4 about our chosen of the GMOS sites among the 43 which constitute to date the network and our reply to the comment n.6 related to the first Referee about the "data coverage". Thank you for your comment.

24) Page 15, Lines 3-11. Here, and in some other places in the document, it seems that you are just restating the information that can be clearly seen in the Figures. The manuscript is pretty long, and perhaps some efficiency could be obtained by omitting at least some of this type of reiteration?

**Reply**: Yes, we agree with you. We tried to optimize/summarize through the manuscript the sentences related to what is clearly showed in the Figures according to. Please, see the revised version of the manuscript as examples at page 13, lines 8-11; page 14, lines 7-11, page 15, lines 1-7 and so on. Thank you.

25) Page 15, Lines 16-17. As mentioned earlier in other contexts, it might be really helpful here to show a graph of flux vs. precipitation amount.

**Reply**: yes, thanks. Please see our previous reply to your comment reported earlier in other context and the revised version of the manuscript and Figures. Thank you once more.

26) Page 17, Line 6. What emissions are larger in the warmer months?

**Reply**: In this part of the paper we want only to point out that during such period of the year, in this case, during the warmer months, the concomitance of meteorological conditions (such as high temperature, higher solar radiations etc.) along with other existing conditions and characterizing a such site (such as local emission sources) could enhance the effects on Hg chemistry that in other situation/conditions (for example during the cold months) not occurs or not occurs with the same "intensity". We reported, in fact, in this part of the manuscript the example of some oxidants (i.e., O3, OH radicals etc.) that under conditions and the photo-oxidation processes are enhanced. These conditions/parameters all together could give enhanced oxidized mercury species in the atmosphere for the conversion of GEM to GOM. In addition, we have made thorough revision in the revised version of the paper to replace "was likely" with "could be" being the first an inaccurate word which could give unclear the sentences (i.e., page 16, line 6). Please, see the revised version of the manuscript. Thank you for your comment.

27) Page 19, Lines 2-3. Perhaps too much to ask, but would it be possible to show maps of emissions in relation to the sites?

**Reply**: We thanks the reviewer for his effort and several input and suggestion that we try to strictly follow because we strongly believe that they are improving the quality of the paper. Anyway, as the reviewer reported above in one of his comment, the manuscript is pretty long, although we tried to synthesize and when it was possible delete several sentences, therefore, considering also the number of Figures and new Tables included partially in a supplementary material document added to the manuscript with further and useful information, we prefer not added additional map or figures. This also considering that some comments reported in the paper on this issue are related to previous studies already published in the literature, therefore, we think that inserting the references in the text this could be sufficient to understand the discussion reported in our manuscript. In addition, we would like to point out that in the same special issue there are other some papers (i.e., De Simone et al.) where modeling application and measurements performed at some of the same GMOS sites are discussed along with useful emission maps. Thank you once more.

28) Page 21, Lines 1-2. Sorry to be repetitive, but again, could may be show a graph of flux vs. precipitation amount.

**Reply**: To accommodate the above comments of this reviewer, we have revised the Figures as also reported earlier in our replies to the comments on this issue. Please see our previous reply and the revised version of the manuscript. Thank you.

29) Page 22, Lines 10-15. Not sure what you mean by "washout". Are you referring to belowcloud scavenging of PBM by falling precipitation? Perhaps you could explain a bit more about the phenomena that you are describing here. **Reply**: yes, we refer to the below-cloud scavenging of PBM by falling precipitation. Following your suggestion, we explain a little bit about the phenomena. Please, see the revised version of the manuscript at page 18, lines 8-11. Thank you.

30) Page 26, Lines 1-3. Here, and at a few other points, you note patterns in relation to GOM or other measurements. Would a graphic be useful here to show the relationship, e.g., GOM vs. volume-weighted-mean concentration in precip?

**Reply**: Thank you for your suggestion. In the original version of the manuscript, we didn't include additional graphic and or details on Hg measurements in air to avoid to report and/or repeat some issue and results reported within other recent manuscripts and/or included within the same special issue (such as Sprovieri et al. where atmospheric Hg results obtained within the GMOS network have been discussed and presented), even if has been often highlighted the need of additional investigation on the relationship of atmospheric Hg speciation measurements vs Hg in precipitation. Anyway, in the revised version of the manuscript, we rewrote and/or deleted some sentences according to the new calculation performed for rainfall amounts and wet deposition fluxes normalizing the weighted data on each sample at each site with 2-weeks reference time as described earlier. The new analysis for AMS data samples gave us different results on the seasonal trend of the precipitation amounts and THg wet deposition flux patterns. These findings make the previous instance on the relationship between GOM vs Hg precipitation data no longer valid. Therefore, we delete the sentence related to the possible correspondence in this case among atmospheric and precipitation data. We are grateful to the reviewer for this and for his previous suggestions which have prompted us to make a more accurate and scientifically objective analysis than that previously made taking into account only the number of sampling days. Please, see the revised version of the manuscript at page 24, lines 9-13, and page 25, lines 1-5. Thank you.

31) Page 27, Lines 30-31. You state that early models tended to overestimate the influence of local emissions sources. This may or may not have been true, for one or more models, but I feel you'd need to cite a lot of different papers really make this statement. To me, seems like an overly provocative statement, and one that is not really needed for the paper? The general idea that observations are critical for model evaluation is certainly valid, but I don't think you can (or need to) make this sweeping statement about "early models". Indeed, Sunderland et al (2016) have recently pointed out that "early models" may have significantly underestimated the influence of local emissions sources! Sunderland, E. M., C. T. Driscoll, J. K. Hammitt, P. Grandjean, J. S. Evans, J. D. Blum, C. Y. Chen, D. C. Evers, D. A. Jaffe, R. P. Mason, S. Goho and W. Jacobs (2016). Benefits of Regulating Hazardous Air Pollutants from Coal and Oil Fired Utilities in the United States. Environmental Science & Technology 50(5): 2117-2120.

**Reply**: Thank you for your comment. We absolutely agree with you regarding the sweeping statement reported in this part of the paper. We revised this section taking into account what recently Sunderland et al. (2016) pointed out. Please, see the revised version of the manuscript at page 27, lines 22-24 within the Section "Conclusions". Thank you once more.

32) Page 27, Line 31. Is this really all available GMOS wet dep data, or just the data from selected sites for selected years? Also, are the GMOS wet dep data (and other data?) available? Perhaps this could be mentioned?

**Reply**: yes, we agree with you, thank you. In this paper we referred only to selected monitoring sites which provided Hg data in precipitation during the development of the GMOS project. Please, see the revised sentence in the new text at page 27, lines 25-28.

However, we would like to point out that Hg measurements across the GMOS network are ongoing, including wet deposition samples, thus the number of the ground-based sites is growing, and the data from them as well. The data coming from the GMOS network are available upon request and protected by a policy document available for the scientific community (i.e., it can be downloaded GMOS (www.gmos.eu), following from the web page please, see the link: http://www.gmos.eu/public/GMOS-Governance Data Policy rev160705.pdf).

33) Page 28, Line 1. Having data a "remote" sites with few local or regional sources is important, for sure, but having data a sites that are influenced by local and regional sources are also important for better understanding of Hg atmospheric fate and transport (and model evaluation), etc.

**Reply**: Yes, thank you, we agree with you. We revised the statement according to. Please, see at page 28, lines 8-10. Thank you.

**Technical Corrections**

1) Page 3, Line 1: ... long-term measurements of ambient Hg concentrations and measurements of Hg wet deposition fluxes were lacking...

Reply: Yes, corrected, thank you. See at page 3, lines 6-8.

2) Table 2. There is a vertical line in the top of the table (see clip below, with red circle), that I think should be removed.

**Reply:** Yes, corrected, thank you. Please, see the supplementary material added to the manuscript where, now Table 2 becomes Table S1.

|                        |     |                                                   | - 1                                                                                                             | -            |                            |                                              | 10-                                               |                  |              |                            |                                              |
|------------------------|-----|---------------------------------------------------|-----------------------------------------------------------------------------------------------------------------|--------------|----------------------------|----------------------------------------------|---------------------------------------------------|------------------|--------------|----------------------------|----------------------------------------------|
|                        |     |                                                   | C                                                                                                               |              | 2011                       |                                              |                                                   |                  | 2012         |                            |                                              |
|                        |     | Annual
Wet Dep. Flux
$[\mu gm^{-2}yr^{-1}]$ | Rainfall
[mm]                                                                                                | ndays
[d] | Weighted
HgT
[ngL-1] | Aver. Wet
Dep. Flux
$[ngm^{-2}d^{-1}]$ | Annual Wet
Dep. Flux
$[\mu gm^{-2}yr^{-1}]$ | Rainfall
[mm] | ndays
[d] | Weighted
HgT
[ngL-1] | Aver. Wet
Dep. Flux
$[ngm^{-2}d^{-1}]$ |
| Northern Hemisphere    | NYA | 15 T                                              | ~                                                                                                               |              | .                   | 121                                          | 0,9                                               | 238,6            | 350          | 3,8                        | 2,6                                          |
|                        | PAL | 2,9                                               | 407,4                                                                                                           | 363          | 7,1                        | 8,0                                          | 1,9                                               | 278,6            | 332          | 6,8                        | 5,7                                          |
|                        | RAO | 5.8                                               | 646.6                                                                                                           | 364          | 8.9                        | 15.8                                         | 6.5                                               | 621.8            | 366          | 10.4                       | 17.8                                         |
|                        | MHE |                                                   | -                                                                                                               |              |                            |                                              | 0,9                                               | 393,7            | 113          | 2,2                        | 7,6                                          |
|                        | LIS | 2.                                                | 100                                                                                                             | i i i        | ŝ.                         | 2.                                           | 0,2                                               | 17,4             | 18           | 9,7                        | 9,4                                          |
|                        | CMA | 84200 B                                           | 1.00%                                                                                                           | ~            | ~                          | 2542.01                                      | ~                                                 |                  |              | 10                         | 200                                          |
|                        | ISK | 5,1                                               | 680,2                                                                                                           | 224          | 7,5                        | 22,7                                         | 8,4                                               | 1349,7           | 363          | 6,2                        | 23,2                                         |
|                        | MCH | 2,8                                               | 264,6                                                                                                           | 119          | 10,6                       | 23,6                                         | 4,8                                               | 569,4            | 228          | 8,4                        | 21,1                                         |
|                        | LON |                                                   | -                                                                                                               | -            | -                          | -                                            | 0,3                                               | 88,2             | 19           | 3,9                        | 18,1                                         |
|                        | MWA |                                                   | -                                                                                                               | - s          | -                          | 100                                          | 0,3                                               | 79,5             | 127          | 4,3                        | 2,7                                          |
|                        | MAL | 4,3                                               | 1543,2                                                                                                          | 222          | 2,8                        | 19,5                                         | 3,2                                               | 971,5            | 202          | 3,3                        | 16,1                                         |
| Tropics                | SIS |                                                   | 1940 - 1940 - 1940 - 1940 - 1940 - 1940 - 1940 - 1940 - 1940 - 1940 - 1940 - 1940 - 1940 - 1940 - 1940 - 1940 - | -            | 8                          | 10 C                                         | 2                                                 | -                |              | -                          | -                                            |
|                        | CST | -                                                 |                                                                                                                 |              | -                          | -                                            | 2,4                                               | 297,1            | 155          | 8,1                        | 15,5                                         |
| Southern
Hemisphere | AMS |                                                   |                                                                                                                 |              | e                          |                                              |                                                   |                  |              | -                          | =                                            |
|                        | CPT | 0,3                                               | 133,5                                                                                                           | 119          | 2,1                        | 2,4                                          | 3.8                                               | 260,3            | 147          | 14,6                       | 25,8                                         |
|                        | CGR |                                                   |                                                                                                                 |              | -                          | 1                                            |                                                   | -                |              | -                          | -                                            |
|                        | BAR | -                                                 | -                                                                                                               |              | -                          | -                                            |                                                   | -                |              | -                          | -                                            |

3) Table 3. I think "uom" refers to "units of measurement", but maybe clearer just to put the units, or spell out "units of measurement. Better yet to include the units directly in the table.

**Reply:** Yes, we followed your suggestion including the units directly in the revised Table 3 (now Table S2 in the supplement material document). Please, see Table S2. Thank you.

4) Page 6, Lines 31-32: ...the number of the sampling days as well as the annual wet deposition flux and average THg wet deposition flux calculated for each year in the period 2011-2015.

**Reply:** Thank you, the sentence has been corrected according to and integrated on the basis of new calculation performed. Please, see at page 6, lines 18-22.

- 5) Page 6, Line 32: As noted above, it is really unclear how valid any of the partial-year data are, given that it is unclear if the missing data are from rainy or dry seasons, etc.
- **Reply:** Please, see our reply on this issue to your comment reported above within the "Specific Comments" section and the revised version of the Section 3 and 4 of the manuscript. Thank you for your input and suggestion.
  - 6) Page 8, Line 17. ... during the 2011-2015 period are is reported in Figure 3

**Reply:** Yes, that's right. However, this sentence in the manuscript has been changed. Please, the revised version at page 8, line 26-28. Thank you.

7) Page 11, Lines 19-24. Seem like sometimes you refer to sites using the 3-letter abbreviation, and sometimes you refer the sites using the full name of the site. Since the graphics all use the 3-letter abbreviation, maybe better to just use these in the text throughout. Could give the full name the first time it was mentioned, with the abbrev in parentheses, and then just use the abbreviation from then on?

**Reply:** Yes, thank you. We followed your suggestion. Please see the revised version of the manuscript.

8) Page 13, Line 13. "meteorological" is misspelled.

**Reply:** Corrected. Please, see at page 12, line 22. Thank you.

9) Page 20, Line 5. "rain" not "rainy"

Reply: Corrected, thank you. Please, see at page 17, line 4 of the revised manuscript.

10) Page 22, Line 5. ... The positive or negative correlation between THg concentrations and the precipitation amount has not been obviously observed at MAL where the rainy samples shows a fairly seasonal variability, during all seasons with lowest average rainfall in winter and the highest in fall...

**Reply:** The sentence has been integrated and **c**orrected, thank you. Please, see at page 17, line 14 and page 18, lines 1-4 of the revised manuscript.

11) Page 23, Line 18. ... exhibit a seasonality in annual rainfall, ...

Reply: Corrected, thank you. Please, see at page 20, line 30 of the revised manuscript.

12) Page 24, Line 10. "fuels" not "flues"

- Reply: Corrected, thank you. Please, see at page 21, line 6 of the revised manuscript.13) Page 24, Line 12. "United States"
- **Reply:** Corrected, thank you. Please, see at page 22, line 2 of the revised manuscript. 14) Page 24, Line 12. "waste incinerators"
- Reply: Corrected, thank you. Please, see at page 22, line 3 of the revised manuscript.

---

## Author Comment (AC3) · 15 Nov 2016

Dear Referee Dr. Cohen,

First of all, we would like to thank you for the effort and useful suggestions that you have done for the manuscript on mercury wet deposition flux performed at the GMOS sites distributed worldwide. We completed the revision of the manuscript according to comments provided by you and the other reviewers taking into account the important input and corrections you highlighted. We appreciate very much the valuable comments for improving the readability and interpretation of the manuscript. We think that after this review, our manuscript has been now improved. Attached to this text, as supplement file we reported point by point our detailed responses to the

comments for each Reviewer. Please, see the pdf file provided and our response to your comments and suggestions. Thank you very much once more.

Please also note the supplement to this comment:
http://www.atmos-chem-phys-discuss.net/acp-2016-517/acp-2016-517-AC3-supplement.pdf

---

## Referee Report (RR1)

Review of MS No.: acp-2016-517: "Five-year records of Total Mercury Deposition flux at GMOS sites in the Northern and Southern Hemispheres", by F. Sprovieri et al.

(2nd Review by Mark Cohen, Dec 10, 2016)

**General Comments**

Francesca Sprovieri and co-authors have prepared a new draft of the manuscript in which they have attempted to respond, as appropriate, to the concerns and suggestions of the three reviewers. In this 2nd review, I will primarily focus on the responses to items mentioned in my first review. In general, the authors appear to have comprehensively responded in thoughtful and constructive ways to my earlier comments, both in the point-by-point "narrative" response and in the changes made to the manuscript. This began as an important, well-written paper, representing a tremendous amount of work, and the authors' have quite impressively improved the paper significantly. There are just a few relatively minor issues remaining that I would like to mention. None of these are "deal-breakers", but the authors may wish to consider them in preparing the final version of the paper.

**Specific Comments**

● Perhaps the wet deposition flux calculation could be explained a little better (e.g., Page 6, lines 4-14). It still seems somewhat unclear to me. Given how fundamental these flux estimates are to the paper, I believe the paper could be improved by additional clarification and/or explanation.

   o First, the text makes it seem like the individual "$P^i$" amounts are the precipitation amounts that occurred during each particular sample. However, based on the calculations I can infer from Tables S1 and S2, it seems that the annual wet dep flux for a given site is being calculated by multiplying the volume-weighted concentration estimated for days that the sampler was operated by the total precipitation continuously measured at the site over the entire year, i.e., including periods when the mercury sampler was not operating. If this is the case, then it should be made clear in the text and in the explanation of Tables S1 and S2. If this is not the case, and the Rainfall (mm) amounts in Tables S1 and S2 only represent the rainfall that was measured during the times that the Hg sampler was operating, then I don't see how an estimate for the entire year could be made.

   o Second, the authors have added in text mentioning that the samples have been "normalized" with respect to a 15-day sampling time (in response to review questions about this issue). It's not clear what is meant by this normalization. If it is too unwieldy to put in the manuscript text, the explanation could be added to the Supplement. I see in some later explanation that this new estimation methodology has changed some of the conclusions (e.g., in relation to GOM vs. wet deposition flux), and so all the more important to make sure it is very clear what this normalization process entailed. This will be useful for future work in the field, as this issue will probably continue to be something that has to be factored into the data analysis.

● Figure 2:

   o Perhaps I have missed it in the text, but I believe it would be helpful to clarify in the caption or the text that the Rainfall (mm) is the total over the entire year, measured continuously, and not just the total for the days that the mercury sampler operated. This is really the same question that is mentioned in the earlier specific comment.

   o How is frozen precipitation represented in these "Rainfall (mm)" values? An explanation of this could be added somewhere in the paper. Should the y-axis be labeled "Precipitation (mm)"? The term "Rainfall" is used throughout the document, but it seems unlikely that all of the precipitation at all of the sites would be liquid "rain". I imagine that the frozen precipitation is melted for each sample and the totals expressed as "liquid rain equivalent".

   o In relation to these points, you wouldn't necessarily have to change any of the Figures, Tables, or text, but perhaps you could add somewhere that "rainfall" and "precipitation" are used interchangeably throughout the paper, and the term "rainfall" includes all forms of precipitation. And you could also mention at some point – if this is indeed true -- that unless otherwise indicated, any "rainfall" or "precipitation" amount is the total measured continuously at a site, independent of whether the mercury sampler was operating at any given time at a site.

**Technical Corrections and Suggestions**

● Capitalization of words in the title seems inconsistent, e.g., "…Mercury wet Deposition flux…". Could change to be consistent with whatever title capitalization conventions used are generally used in ACP.

● Page 2, lines 8-12: New sentence regarding dry deposition seems to be too long, and could potentially be reworded as follows:

   "Currently, Hg dry deposition is often estimated by models, using measured ambient concentrations of Hg measurements and meteorological parameters, due to the lack of existing direct and accurate measurement methodologiess (Gustin et al., 2012; Zhang et al., 2012).; Ttherefore the investigation of Hg fluxes to terrestrial and aquatic surfaces in different parts of the world are oftenmainly based on performed by wet deposition measurements (Gratz et al., 2009; Feng et al., 2009)."

● Reference in new sentence about IVL-Bulk sampler unclear (page 4, line 26):  "en, 2010". Is "en" the last name of the author?

● Tables S1 and S2 – it is stated that "Measures in bold are related to the calculations based on a restricted number of sampling days, therefore statistically less representative than the others". However, I don't seem to see any bold vs. non-bold entries. If you do differentiate the entries as to significance, then expressing the less significant values as "bold" seems a little counterintuitive.

---

## Author Response (AR2)

[revised manuscript text omitted]

(2nd Review by Mark Cohen, Dec 10, 2016)

First of all, we appreciate very much the effort and the useful suggestions reported for the manuscript on mercury wet deposition flux performed at the GMOS sites distributed worldwide done by Dr. Cohen. We followed the suggestion and revised the manuscript accordingly. Below we reported point by point our responses to the comments/suggestion of Dr. Cohen.

Thank you very much once more.

**Specific Comments**

● Perhaps the wet deposition flux calculation could be explained a little better (e.g., Page 6, lines 4-14). It still seems somewhat unclear to me. Given how fundamental these flux estimates are to the paper, I believe the paper could be improved by additional clarification and/or explanation.

  o First, the text makes it seem like the individual "$P_i$" amounts are the precipitation amounts that occurred during each particular sample. However, based on the calculations I can infer from Tables S1 and S2, it seems that the annual wet dep flux for a given site is being calculated by multiplying the volume-weighted concentration estimated for days that the sampler was operated by the total precipitation continuously measured at the site over the entire year, i.e., including periods when the mercury sampler was not operating. If this is the case, then it should be made clear in the text and in the explanation of Tables S1 and S2. If this is not the case, and the Rainfall (mm) amounts in Tables S1 and S2 only represent the rainfall that was measured during the times that the Hg sampler was operating, then I don't see how an estimate for the entire year could be made.

  o Second, the authors have added in text mentioning that the samples have been "normalized" with respect to a 15-day sampling time (in response to review questions about this issue). It's not clear what is meant by this normalization. If it is too unwieldy to put in the manuscript text, the explanation could be added to the Supplement. I see in some later explanation that this new estimation methodology has changed some of the conclusions (e.g., in relation to GOM vs. wet deposition flux), and so all the more important to make sure it is very clear what this normalization process entailed. This will be useful for future work in the field, as this issue will probably continue to be something that has to be factored into the data analysis.

● Figure 2:

  o Perhaps I have missed it in the text, but I believe it would be helpful to clarify in the caption or the text that the Rainfall (mm) is the total over the entire year, measured continuously, and not just the total for the days that the mercury sampler operated. This is really the same question that is mentioned in the earlier specific comment.

  o How is frozen precipitation represented in these "Rainfall (mm)" values? An explanation of this could be added somewhere in the paper. Should the y-axis be labeled "Precipitation (mm)"? The term "Rainfall" is used throughout the document, but it seems unlikely that all of the precipitation at all of the sites would be liquid "rain". I imagine that the frozen precipitation is melted for each sample and the totals expressed as "liquid rain equivalent".

o In relation to these points, you wouldn't necessarily have to change any of the Figures, Tables, or text, but perhaps you could add somewhere that "rainfall" and "precipitation" are used interchangeably throughout the paper, and the term "rainfall" includes all forms of precipitation. And you could also mention at some point – if this is indeed true -- that unless otherwise indicated, any "rainfall" or "precipitation" amount is the total measured continuously at a site, independent of whether the mercury sampler was operating at any given time at a site.

**Reply to both "Specific Comments" reported above:**

We thank the reviewer for pointing out this important issue on wet deposition flux calculation. We agree with you him about the need of some further clarification and/or explanation, therefore, we revised the whole paragraph at Page 6 of the manuscript following the input of the Reviewer, and revising the text according to. Please, see the Section 2.3 at Page 6 of the revised version of the manuscript. Thank you very much once more.

**Technical Corrections and Suggestions**

● Capitalization of words in the title seems inconsistent, e.g., "...Mercury wet Deposition flux...". Could change to be consistent with whatever title capitalization conventions used are generally used in ACP.

**Reply:** Yes, Thank you we followed your suggestion. Please, see the new Title now consistent with capitalization conventions. Thank you.

● Page 2, lines 8-12: New sentence regarding dry deposition seems to be too long, and could potentially be reworded as follows:
"Currently, Hg dry deposition is often estimated by models, using measured ambient concentrations of Hg measurements and meteorological parameters, due to the lack of existing direct and accurate measurement methodologies (Gustin et al., 2012; Zhang et al., 2012)., Therefore the investigation of Hg fluxes to terrestrial and aquatic surfaces in different parts of the world are often mainly based on performed by wet deposition measurements (Gratz et al., 2009; Feng et al., 2009)."

**Reply:** Thank you, the sentence has been corrected according to. Please, see at page 2, lines 12-17

● Reference in new sentence about IVL-Bulk sampler unclear (page 4, line 26): "en, 2010". Is "en" the last name of the author?

**Reply:** Thank you, the reference within the sentence has been rewrite to make it more clear. Please, see at page 5, lines 1.

● Tables S1 and S2 – it is stated that "Measures in bold are related to the calculations based on a restricted number of sampling days, therefore statistically less representative than the others". However, I don't seem to see any bold vs. non-bold entries. If you do differentiate the entries as to significance, then expressing the less significant values as "bold" seems a little counterintuitive.

**Reply:** Thank you for highlighting this point. We removed the sentence "Measures in bold are related to the calculations based on a restricted number of sampling days,

therefore statistically less representative than the others" in order to avoid ambiguity. Please, see new Captions of Tables S1 and S2.